# An intersectional gene regulatory strategy defines subclass diversity of *C. elegans* motor neurons

Paschalis Kratsios[1]*, Sze Yen Kerk[2], Catarina Catela[1], Joseph Liang[3], Berta Vidal[2], Emily A Bayer[2], Weidong Feng[1], Estanisla Daniel De La Cruz[2], Laura Croci[4], G Giacomo Consalez[4,5], Kota Mizumoto[3], Oliver Hobert[2]*

[1]Department of Neurobiology, University of Chicago, Chicago, United States; [2]Department of Biological Sciences, Howard Hughes Medical Institute, Columbia University, New York, United States; [3]Department of Zoology, The University of British Columbia, Vancouver, Canada; [4]Division of Neuroscience, San Raffaele Scientific Institute, Milan, Italy; [5]Università Vita-Salute San Raffaele, Milan, Italy

**Abstract** A core principle of nervous system organization is the diversification of neuron classes into subclasses that share large sets of features but differ in select traits. We describe here a molecular mechanism necessary for motor neurons to acquire subclass-specific traits in the nematode *Caenorhabditis elegans*. Cholinergic motor neuron classes of the ventral nerve cord can be subdivided into subclasses along the anterior-posterior (A-P) axis based on synaptic connectivity patterns and molecular features. The conserved COE-type terminal selector UNC-3 not only controls the expression of traits shared by all members of a neuron class, but is also required for subclass-specific traits expressed along the A-P axis. UNC-3, which is not regionally restricted, requires region-specific cofactors in the form of Hox proteins to co-activate subclass-specific effector genes in post-mitotic motor neurons. This intersectional gene regulatory principle for neuronal subclass diversification may be conserved from nematodes to mice.

*For correspondence: pkratsios@ uchicago.edu (PK); or38@ columbia.edu (OH)

**Competing interests:** The authors declare that no competing interests exist.

## Introduction

An obligatory first step toward understanding nervous system development, function and evolution is the cataloguing of its individual building blocks. Such cataloguing involves the classification of neurons into classes and the subdivision of classes into more refined subclasses. Historically, neurons have been grouped into specific classes and subclasses based on anatomical and electrophysiological features. The simplicity of invertebrate nervous systems permitted comprehensive classification schemes based on anatomy alone, as best exemplified by the electron microscopical reconstruction of the *Caenorhabditis elegans* nervous system (*White et al., 1986*). In vertebrates, the advent of comprehensive molecular profiling technology has revolutionized neuronal classification efforts. Recent RNA-sequencing studies on specific neuron classes of the brainstem (*Gaspar and Lillesaar, 2012*; *Jensen et al., 2008*; *Okaty et al., 2015*; *Spaethling et al., 2014*), the midbrain (*Poulin et al., 2016*, *2014*; *Roeper, 2013*), the cortex (*Tasic et al., 2016*; *Zeisel et al., 2015*), the retina (*Macosko et al., 2015*; *Shekhar et al., 2016*) and the spinal cord (*Bikoff et al., 2016*; *Gabitto et al., 2016*) have revealed a recurring theme. Neurons can be grouped together based on a number of shared molecular, functional and anatomical traits, but can be further subdivided into individual subclasses based on subclass-specific traits. For example, serotonergic neurons in the vertebrate central nervous system are a group of neurons in the Raphe nuclei of the brainstem defined by their usage of the same neurotransmitter and their specification by the transcription factor PET1,

but they fall into specific classes based on anatomical, functional and molecular features (*Gaspar and Lillesaar, 2012*; *Jensen et al., 2008*; *Okaty et al., 2015*). These classes can be further subdivided into distinct subclasses based on molecular features (*Okaty et al., 2015*). Similarly, recent studies have revealed substantial diversity of the V1 class of spinal inhibitory neurons that can be subdivided into subclasses based on a number of molecular and electrophysiological features (*Bikoff et al., 2016*; *Gabitto et al., 2016*). Although we have begun to understand the molecular mechanisms that generate individual neuron classes, it remains relatively poorly understood how subclass diversity is genetically programmed, i.e., how neuronal classes are instructed to diversify further into more refined subclasses.

To be able to study the problem of how neuronal subclass diversity is generated, we use the nervous system of the nematode *C. elegans* as a model. The 302 neurons of the nervous system of the hermaphrodite have been classified into 118 anatomically distinct classes (*Hobert et al., 2016*; *White et al., 1986*). Synaptic connectivity patterns as well as molecular markers suggest that these distinct classes can be subdivided into subclasses (*Hobert et al., 2016*). One paradigm for such 'subclass diversification' of individual neuron classes is provided by cholinergic motor neurons (MNs) in the *C. elegans* ventral nerve cord (VNC), retrovesicular ganglion, and preanal ganglion. As shown in *Figure 1A*, these cholinergic MNs can be divided into seven classes, the embryonically generated DA, DB, and SAB, and the post-embryonically generated AS, VA, VB and VC classes (*Von Stetina et al., 2006*; *White et al., 1976*). Each MN class is defined by its unique morphology that is shared by each individual class member (DA = 9 class members, DB = 7, SAB = 3, VA = 12, VB = 11, VC = 6, AS = 11) that intermingle along the A-P axis (*Figure 1A*). Traditionally, each one of these MN classes has been defined by a combination of class-specific axodendritic projection patterns and class specific patterns of synaptic connectivity (*White et al., 1986*). This classification is corroborated by the class-specific expression of a unique combination of effector genes that assign unique features to each MN class (*Hobert et al., 2016*). Since these effector genes are expressed in all individual neurons that belong to a certain MN class, we refer to them as 'MN class-specific genes'. These include genes encoding neurotransmitter receptors, ion channels, gap junction proteins and signaling molecules (*Figure 1B*).

There are three lines of evidence that demonstrate the existence of subclass diversity within each cholinergic MN class. First, the *C. elegans* wiring diagram reveals a correlation of connectivity differences with MN cell body position along the A-P axis and, more specifically, with the location of individual class members in the retrovesicular ganglion, VNC and preanal ganglion. For example, although all nine members of the DA neuron class share multiple connectivity features, such as input from the command (pre-motor) interneurons AVA and AVD and output to dorsal body wall muscle, the DA neurons located in the preanal ganglion (DA8 and DA9) display several synaptic connectivity differences when compared to the more anteriorly located DA neurons (DA1-DA7) which are located in the VNC and retrovesicular ganglion (*Figure 1B*, *Figure 1—source data 1*; www.wormwiring.org). Subclass diversity based on synaptic connectivity is not only evident in the DA class, but it is a recurring theme for all cholinergic MN classes (*White et al., 1986*) (*Figure 1B*, *Figure 1—source data 1*). Second, studies of *C. elegans* mutants provided further evidence for MN subclass diversity. A paired-type homeobox gene, *unc-4*, and a Groucho corepressor, *unc-37*, are selectively required only in VA neurons located in the VNC (VA2-VA10), but not in the retrovesicular ganglion (VA1) or preanal ganglion (VA11,VA12) to specify the correct pattern of presynaptic input from AVA pre-motor neurons (*Miller et al., 1992*; *Pflugrad et al., 1997*; *White et al., 1992*). Third, reporter gene data indicates that individual neurons from a given MN class can be distinguished molecularly (*Figure 1B*). Once again, there is a correlation with cell body position. For example, as shown in *Figure 1B*, the TGFβ-like molecule *unc-129* is selectively expressed in the DA neurons of the retrovesicular ganglion and VNC (DA1-7) and not in DA8 and DA9 neurons in the preanal ganglion (*Colavita et al., 1998*). Conversely, the *itr-1* gene is exclusively expressed in DA9 (*Klassen and Shen, 2007*). Since these effector genes are not expressed in all individual neurons of a given class, we refer to them as 'MN subclass-specific genes'. We identify in this paper a number of novel MN subclass-specific genes and further reveal a mechanism for their transcriptional regulation, providing key insights into how neuronal subclass diversity is generated along the A-P axis.

The functional relevance of subclass diversification becomes apparent if one considers the *C. elegans* synaptic wiring diagram (*White et al., 1986*). While MN class members located in the VNC or the preanal ganglion share many synaptic partners (schematically shown in *Figure 1B* and

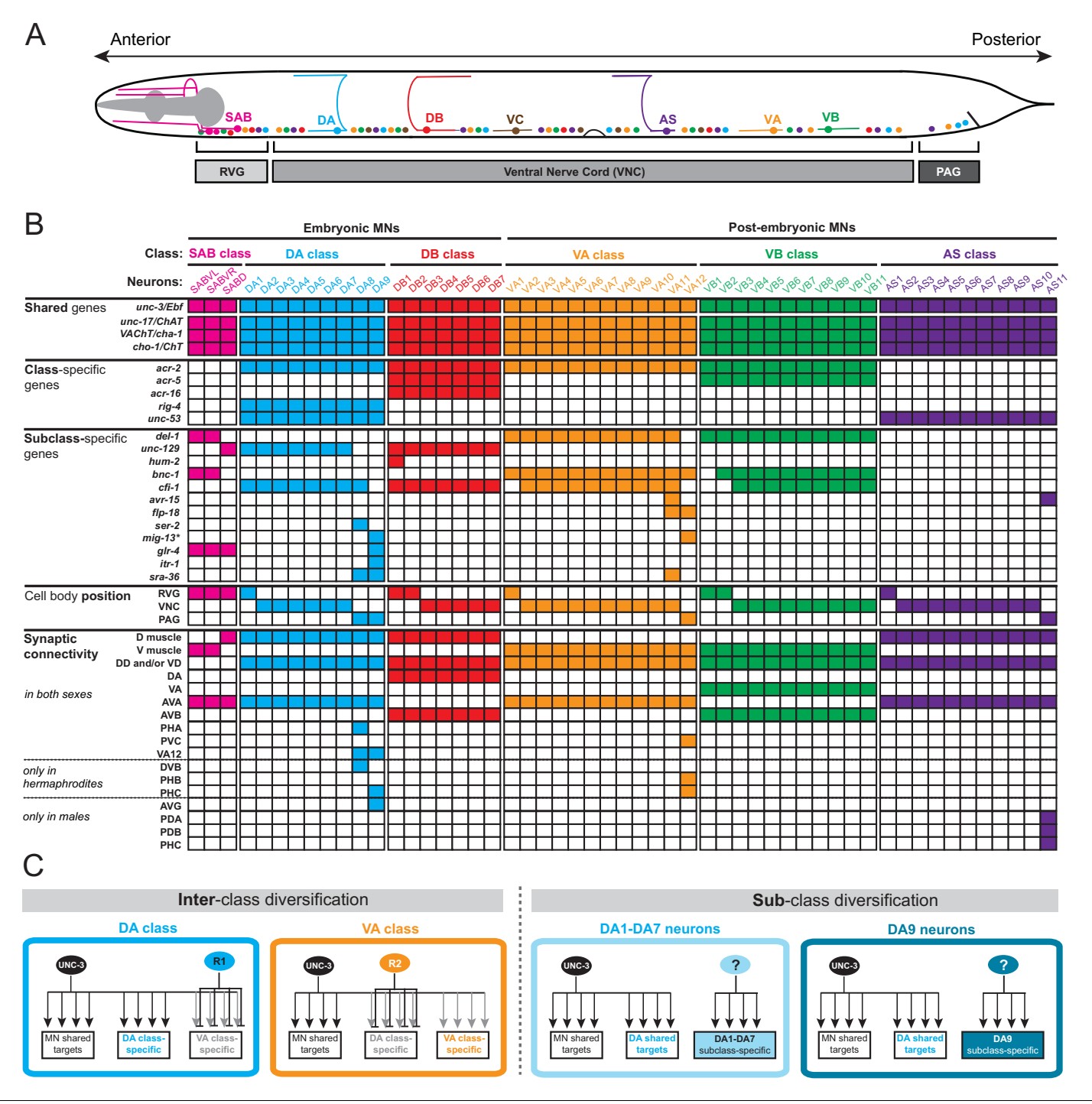

**Figure 1.** A map of subclass-specific genes provides an entry point to study MN subclass diversification. (A) Schematic showing seven *C. elegans* MN classes (SAB, DA, DB, VA, VB, AS, VC), which are color-coded. Individual neurons of each class intermingle and populate three regions along the A-P axis; retrovesicular ganglion (RVG), ventral nerve cord (VNC), preanal ganglion (PAG). Axonal trajectory is shown only for one member of each class. (B) A map of effector gene expression with single-cell resolution. Each column represents an individual neuron. Genes that are shared, class- and subclass-specific are shown on the left. Cell body position and connectivity similarities and differences are also shown for each individual motor neuron. Asterisk next to *mig-13* indicates additional expression (not shown) in MNs located in the VNC but anterior to vulva. (C) Inter-class versus sub-class diversification. The left panel shows known mechanisms for interclass-specification. Class-specific repressor proteins counteract *unc-3*'s activity, thereby generating 'inter'-class diversity in *C. elegans* MNs (example of DA and VA class is shown) (*Kerk et al., 2017*). The right panel illustrates the problem of

*Figure 1 continued on next page*

*Figure 1 continued*

subclass diversification: Within a given class, the mechanisms controlling the expression of subclass-specific genes and thereby generate 'intra'-class diversity are not known (example of DA class is shown).

The following source data and figure supplement are available for figure 1:

**Source data 1.** Summary of synaptic wiring data.

**Figure supplement 1.** Validation of MN subclass-specific gene expression.

summarized in *Figure 1—source data 1*), several of the posterior-most MN class members generate unique electrical and/or chemical synaptic connections (*Figure 1B*, *Figure 1—source data 1*) (*White et al., 1986*). In the tail, several of these subclass-specific synaptic connections are inputs from tail sensory neurons. This selective input presumably enables individual, posterior-most MN class members to integrate information received from the head (via command/pre-motor interneurons that innervate all MN class members) with information from the tail, thereby providing unique control of tail muscle.

Sexual dimorphisms of the *C. elegans* tail are another likely cause for subclass-specific synaptic connectivity patterns of MNs. During sexual maturation in late larval stages, 68 male-specific sensory, inter- and motor neurons are generated exclusively in the male tail (*Sulston et al., 1980*). The posterior-most members of all five cholinergic VNC MN classes make extensive synaptic connections with these male-specific neurons (*Jarrell et al., 2012*) (*Figure 1—source data 1*). Moreover, there are sexual dimorphisms in the synaptic connections that the posterior-most VNC MN subclasses make with neurons that are present in both sexes (*Jarrell et al., 2012*; *White et al., 1986*). For example, the sex-shared AVG interneurons innervate DA9 only in males (*Figure 1B*). Vice-versa, the sex-shared PHB sensory neuron innervates VA12 only in hermaphrodites (*Figure 1B*) (*Jarrell et al., 2012*; *White et al., 1986*). Taken together, while sharing many molecular and synaptic features, motor neuron classes diversify into subclasses based on unique synaptic connectivity patterns that endow individual MN subclasses with the ability to communicate with specific parts of the nervous system, often in a sexually dimorphic manner.

Here, we provide insights into the molecular logic for how this subclass diversity is genetically encoded along the A-P axis. We describe that the terminal selector-type transcription factor UNC-3, which controls the identity of all distinct MN classes (*Kratsios et al., 2015*, *2011*), is also required for the expression of MN subclass-specific features. However, the manner by which UNC-3 activity is restricted to individual subclasses (members of a given class) is fundamentally distinct from the manner by which UNC-3 activity is restricted to distinct MN classes. In regard to the specification of different MN classes, we have previously described that the activity of UNC-3 is counteracted by class-specific repressor proteins, resulting in 'interclass diversity' (*Figure 1C*)(*Kerk et al., 2017*). In contrast, for 'subclass diversification', that is, diversification of subclasses within a given MN class (schematized in *Figure 1C*), we find that UNC-3 requires region-specific cofactors in the form of Hox proteins to co-activate subclass-specific MN features along the A-P axis. Our analysis hence reveals an intersectional strategy in which a non-regionally restricted, MN class selector gene (*unc-3*) cooperates with a region-specific transcriptional code (Hox) to distinguish MN class members from one another, and thereby generate subclass diversity.

## Results

### Motor neuron classes can be divided into subclasses

To study the problem of neuronal subclass diversification (*Figure 1B*), we sought to expand the available set of 'subclass-specific' markers. We undertook a candidate gene approach examining the precise expression pattern of terminal differentiation (effector) genes that code for neurotransmitter receptors, signaling proteins or ion channels that were previously reported to be expressed in MNs (www.wormbase.org) or that we identified in surveys of expression patterns of specific gene families. All of the genes examined are continuously expressed in cholinergic MNs throughout the *C. elegans*

lifespan. Confirming the notion that all neurons of a given class share a number of molecular features, we observed that several of these MN-expressed genes (the neuronal IgCAM molecule *rig-4*, the acetylcholine receptor subunits *acr-2, acr-5, acr-16*, and *unc-53*, the *C. elegans* ortholog of the human neuron navigator genes Nav-1/2/3) are expressed in all individual members of a given class. We call these 'MN class-specific' markers (*Figure 1B*). In addition, we identified several 'MN subclass-specific' markers, that is, terminal differentiation genes that are continuously expressed in subsets of neurons of a given class (*Figure 1B*, *Figure 1—figure supplement 1*). Some of these genes have been previously described, but their expression pattern was either incomplete or assigned to the incorrect subclass (*Table 1*).

In total, twelve MN subclass-specific markers (most of them are neurotransmitter receptors, ion channels or neuropeptides) molecularly subdivide each MN class (*Figure 1B*, *Table 1*). This molecular subdivision is strictly correlated with cell body position in specific ganglia (*Figure 1B*). For example, of the nine DA neurons, seven are located in the retrovesicular ganglion and VNC (DA1-7), while the remaining two, DA8 and DA9, are located in the preanal ganglion. We identified molecular markers that differentiate the DA MNs in the VNC from those in the preanal ganglion (*Figure 1B*). Similarly, the VA12 neuron, the only VA neuron located in the preanal ganglion, can be molecularly distinguished from more anteriorly located VA1-VA11 (*Figure 1B*, *Figure 1—figure supplement 1*). Lastly, the AS11 neuron, the only AS neuron located in the preanal ganglion, appears to also be molecularly distinct from the AS1 to AS10 neurons that are located more anteriorly (*Figure 1B*, *Figure 1—figure supplement 1*). These subclass-specific markers provide us with an entry point to study the problem of neuronal subclass diversity with single cell resolution.

The molecular subdivision of MN classes into individual subclasses correlates not only with position in ganglia, but also with synaptic connectivity (*White et al., 1986*). The most posterior members of the DA, VA, and AS MN classes located in the preanal ganglion (DA8, DA9, VA12, AS11) display notable connectivity differences when compared to their VNC counterparts (*Figure 1B*, *Figure 1—source data 1*). For example, DA9 in hermaphrodites, unlike any other DA neuron, is innervated by the glutamatergic neuron PHC. Our molecular map revealed that *glr-4*, which codes for an ionotropic glutamate receptor (GluR) subunit, is only expressed in DA9 (*Figure 1B*, *Figure 1—figure supplement 1*), suggesting that this selective *glr-4* expression enables glutamatergic input and thereby provides DA9 with subclass-specific traits. Similarly, AS11, the most posterior AS neuron located in the preanal ganglion, is post-synaptic to the glutamatergic neuron PHC in males, and *avr-15*, a glutamate-gated chloride channel subunit, is selectively expressed in the posterior AS11 neuron (*Figure 1B – Figure 1—figure supplement 1*). A correlation of connectivity and molecular differences was also observed for the most anterior members of each MN class located in the retrovesicular ganglion (*Figure 1—source data 1*). To sum up, the molecular subdivision of MN classes into subclasses is an indicator for subclass connectivity differences.

## The terminal selector gene *unc-3* is required for MN subclass diversity

The conserved COE-type terminal selector *unc-3* is expressed in every neuron that belongs to the DA, DB, VA, VB, and AS class of cholinergic MNs. It controls shared features (e.g. ACh pathway genes) by all MN classes, as well as class-specific features, that are shared by each member of an individual MN class (*Kratsios et al., 2011*). We asked whether UNC-3 also controls subclass-specific genes (*Figure 1B*). We indeed found that *unc-3* regulates the subclass-specific *mig-13, itr-1* and *glr-4* markers that are specifically expressed in the most posterior MN of the DA class located in the preanal ganglion, DA9 (*Figure 2A–B*).

At the mechanistic level, UNC-3 can either control the expression of DA9 subclass-specific genes directly or through an intermediate regulatory factor. To distinguish between these possibilities, we surveyed the *cis*-regulatory regions of the DA9-expressed genes and found UNC-3 binding sites (termed 'COE motifs' [*Kratsios et al., 2011*]), suggesting direct regulation of these MN subclass-specific genes by UNC-3. Small pieces of elements upstream of *glr-4* and *mig-13* that contain the COE motif are sufficient to drive reporter gene expression in DA9 and mutational analysis (*glr-4 538 bp* $^{COE\ MUT}$) shows that the COE motif is essential for this expression (*Figure 2D–E*). Similarly, UNC-3 directly controls the expression of *unc-129*/TGF$\beta$, which is expressed in more anteriorly located DA neurons, namely the DA1-7 neurons (*Figure 2C* and [*Kratsios et al., 2011*]). Apart from the DA neuron class, *unc-3* is also required to regulate subclass-specific genes in the VA neurons, namely the expression of the VA12-expressed genes *mig-13* and *flp-18*, and of *del-1*, which is expressed in

**Table 1.** Motor neuron subclass markers.

| Gene | Function | Previously reported subclass expression | Confirmed subclass expression | Revised/new subclass expression | Stage for onset of expression (in hermaphrodites) | Sex-specificity of expression | UNC-3 dependency and COE motif number |
|---|---|---|---|---|---|---|---|
| del-1 | DEG/ENaC ion channel | VA1-VA12 VB1-VB11 SABVL/R* | | VA1-VA11 VB1-VB11 SABVL/R† | VA1-VA11: late L2‡ VB1-VB11: early L2 SABVL/R: 3-fold embryo | Yes, in males VA12 also express del-1 | Yes (3 motifs) |
| unc-129 | TGF-beta like molecule | DA1-DA7, DB1-DB7, SABD§ | DA1-DA7, DB1-DB7, SABD# | | DA1-DA7, DB1-DB7, SABD: L1 | No | Yes (3 motifs) |
| hum-2 | unconventional myosin | DB1¶ | DB1** | | DB1: L1 or earlier | No | N. D (1 motif) |
| bnc-1 | C2H2 Zn finger transcription factor | VA1-VA12, VB2-VB11, SABVL/R†† | VA1-VA12, VB2-VB11, SABVL/R‡‡ | | VA1-VA12, VB2-VB11, SABVL/R: late L1 | Yes, CP7 and CP8†† in male VNC | Yes[7] (4 motifs) |
| cfi-1 | ARID-type transcription factor | DA1-DA8§§, DB1-DB7 | DA1-DA8##, DB1-DB7 | VA2-VA11, VB3-VB11## | DA1-DA8: not determined VA2-11, VB3-VB11: late L1 | Not determined | Yes (>5 motifs) |
| avr-15 | Glutamate-gated cloride channel | DA9, VA12¶¶ | | VA11, AS11*** | VA11, AS11: late L3 | Yes, 4 additional neurons (not ACh) in male PAG | Yes (4 motif) |
| flp-18 | FMRF-like neuropeptide | none | | VA11 VA12††† | VA11: L2/L3 VA12: L2/L3 | Yes, 8 additional cells/neurons in male tail | Yes (1 motif) |
| ser-2 | tyramine receptor | DA9‡‡‡ | | DA8§§§ | DA8: L1 or earlier | No | No (0 motifs) |
| mig-13 | transmembrane protein | DA9, VA12### | DA9 VA12¶¶¶ | | DA9: L1 or earlier VA12: L2 | Yes, 2 additional ACh tail neurons | Yes (1 motif) |
| glr-4 | Glutamate receptor subunit | none | | DA9**** | DA9: L1 or earlier | No | Yes (2 motifs) |
| itr-1 | Inositol triphosphate receptor | PDA, DA9†††† | | DA9†††† | DA9: L2/L3 | No | Yes (1 motif) |
| sra-36 | G-protein coupled receptor | none | | DA8 DA9 VA11‡‡‡‡ | DA8, DA9: L1 or earlier VA11: L2 | No | N. D (2 motifs) |

*Reagent: *wdIs3 X* or *wdIs6 II [del-1prom::gfp]*; **Winnier et al., 1999**.

†Expression was confirmed in VA1-11, VB1-11, and SABVL/R based on cell body position, axonal trajectory, and expected number of neurons for each class. Also, see **Figure 2C**.

‡As described in **Winnier at al. (1999)**, by the end of L2, *del-1prom::gfp* is visible in a few anterior VAs. This expression progresses in a wave from anterior to posterior VAs, with all VAs (except VA12) expressing *del-1prom::gfp* by the L4/adult stage.

§Reagent: *evIs82B IV [unc-129prom::gfp]*; **Colavita et al. (1998)**, **Kratsios et al. (2015)**

#Expression was confirmed in DA1-7, DB1-7, and SABD based on cell body position, axonal trajectory, cell type-specific markers, and expected number of neurons for each class. Also, see **Figure 2C**.

¶Reagent: *mdIs123 [hum-2prom::gfp]* from James Rand and Stephen Fields; www.wormatlas.org

**Expression was confirmed in DB1 based on cell body position and axonal trajectory. Also, see **Figure 1B**.

††Reagent: *bnc-1 (ot845[bnc-1::+mNG+AID])* CRISPR allele; **Kerk et al., 2017**.

‡‡Expression was confirmed in VA1-VA12, VB2-VB11, SABVL/R based on cell body position and expected number of neurons for each class.

§§Reagent: *otEx6502 [cfi-1fosmid::gfp]*; **Kerk et al., 2017**.

##Expression was confirmed in DA1-8 based on cell body position and the cell-type specific markers *juIs14[acr-2::gfp]* that labels DA, DB, VA, VB. This analysis also revealed that *cfi-1* is not expressed in VA12. No expression in VA1, VB1 and VB2 was inferred based on cell body position and total number (3) of MNs expressing *cfi-1* at the retrovesicular ganglion.

¶¶Reagent: *adEx1299[avr-15prom::gfp]*; **Dent et al., 1997**.

***Expression was confirmed in AS11 based on cell body position, axonal trajectory, and cell type-specific markers. Also, see Figure 9D–E and **Figure 1-figure supplement 1**.

†††Reagent: *ynIs59[flp-18prom::gfp]* from Chris Li. Expression in VA11 and VA12 was confirmed with cell type-specific markers. Also, see **Figure 1B**, **Figure 2A–B**, and **Figure 1—figure supplement 1**.

‡‡‡Reagent: *otIs107 [ser-2^prom1::gfp]*; **Tsalik et al., 2003**.

§§§Expression in DA8 was confirmed with cell type-specific markers. Also, see **Figure 1B** and **Figure 1—figure supplement 1**.

###Reagent: *muIs42[mig-13^prom::MIG-13::gfp]*; **Sym et al., 1999**, **Klassen and Shen (2007)**

¶¶¶Expression in DA9 and VA12 was confirmed with cell type-specific markers. Also, see **Figure 2A–B**.

****Reagent: *otIs476[glr-4^prom::tagrfp]*. Expression in DA9 was confirmed with cell type-specific markers. Also, see **Figure 1—figure supplement 1**.

††††Reagent: *otIs453 [itr-1^prom::gfp]*. Plasmid used for otIs453 transgene is pBT001, which was generated by **Gower et al., 2001** and *gfp* expression in *C.elegans* in DA9 and PDA is described in **Gower et al., 2001** and **Klassen and Shen (2007)**. Expression was confirmed in DA9 using cell type-specific markers. Also, see **Figure 2A–B** and **Figure 1—figure supplement 1**.

‡‡‡‡Reagent: *sEx11976[sra-36^prom::gfp]*. Expression in VA11, DA8 and DA9 was confirmed with cell type-specific markers. Also, see **Figure 1B** and **Figure 1—figure supplement 1**. †, #, **, ‡‡, ##, ¶¶, †††, §§§, ###, ****, ††††See also 'Motor Neuron Subclass Identification' section in Materials and methods.

N. D: Not determined.

VA1-VA11 neurons (**Figure 2B–C** and [**Kratsios et al., 2011**]). Small elements containing a single COE motif upstream of *mig-13* are sufficient to drive reporter gene expression in VA12 (**Figure 2E**). Within the AS neuron class, we find that *unc-3* is also required for the expression of *avr-15/GluR*, a subclass-specific gene expressed in the most posterior AS neuron, AS11 (see further below). We conclude that UNC-3 is required for subclass diversity in DA, VA and AS neurons by controlling directly the expression of subclass-specific genes, as evidenced by the overrepresentation of COE motifs in the *cis*-regulatory region of these genes (**Table 1**).

Since *unc-3* is expressed in all nine neurons of the DA class and all twelve neurons of the VA class (**Kratsios et al., 2011**; **Pereira et al., 2015**), the question arises why UNC-3 does not activate the subclass-specific genes in all members of a certain class? In principle, one can envision the existence of subclass-specific repressor proteins that counteract the ability of UNC-3 protein to activate sub-class-specific genes; only in subclasses lacking such repressors, UNC-3 would be able to activate subclass-specific genes. Such a repressor mechanism has been shown to explain how 'interclass' diversity is generated, i.e. how different neuron classes (e.g. DA vs. VA or DB vs. VB) are generated in *C. elegans* (**Kerk et al., 2017**). (**Figure 1C**). In this case, class-specific repressor proteins constrain the ability of UNC-3 to activate, for example, DA-specific genes in the VA neurons (**Kerk et al., 2017** )(**Figure 1C**). As we will describe below, we found evidence of an alternative, co-activation mechanism operating in MN subclasses to define 'subclass' diversification.

### The *C. elegans* Hox genes *egl-5*/Abd-B/Hox9-Hox13, *lin-39*/Scr/Dfd/ Hox4-Hox5 and *mab-5*/Antp/Hox6-Hox8 are expressed in a MN subclass-specific manner

The spatial segregation of MN subclasses (as defined by the molecular and connectivity criteria described above) in distinct anatomical regions along the A-P axis (retrovesicular ganglion, VNC, preanal ganglion) led us to investigate whether region-specific regulatory factors cooperate with UNC-3 to control the expression of MN subclass-specific features. Hox genes were previously shown to be expressed in spatially distinct domains of the *C. elegans* body (**Kenyon, 1986**; **Kenyon and Wang, 1991**; **Kenyon et al., 1997**). We pursued this lead using fosmid-based reporters (fosmids are large genomic clones of ~30 kb that contain all upstream and downstream *cis*-regulatory elements required for expression of a gene of interest) and found that three out of the six *C. elegans* Hox genes are differentially expressed in subsets of MNs that belong to the same class (**Figure 3A–D**). Specifically, *egl-5*/Abd-B/Hox9-Hox13 is expressed in the DA9 neuron, while *mab-5*/Antp/Hox6-Hox8 is expressed in the DA8 neuron as evidenced by our expression analysis using DA8- and DA9-specifc *rfp* markers (**Figure 3C**). We found no expression of *lin-39/Dfd/*Hox4-Hox5 (nor of *nob-1* and *php-3*, the two posterior-most Hox genes) in DA8 and/or DA9 neurons. However, *lin-39/Dfd/*Hox4-Hox5 and *mab-5/Antp*/Hox6-Hox8 are expressed in more anteriorly located DA neurons (**Figure 3B**). Within the VA class, we found that the posterior Hox gene *egl-5* is expressed in VA12 (located in the preanal ganglion), while *lin-39* and *mab-5* are expressed in the more anteriorly located VA neurons along the VNC (**Figure 3B–C**). Within the AS class, *mab-5* is expressed in the most posterior AS neuron, AS11, the only AS neuron in the preanal ganglion, while its expression overlaps with *lin-39* in more anterior AS neurons located along the VNC (**Figure 3B–C**). Neurons of the other two classes (DB and VB) located in the VNC (but not the preanal ganglion) also coexpress

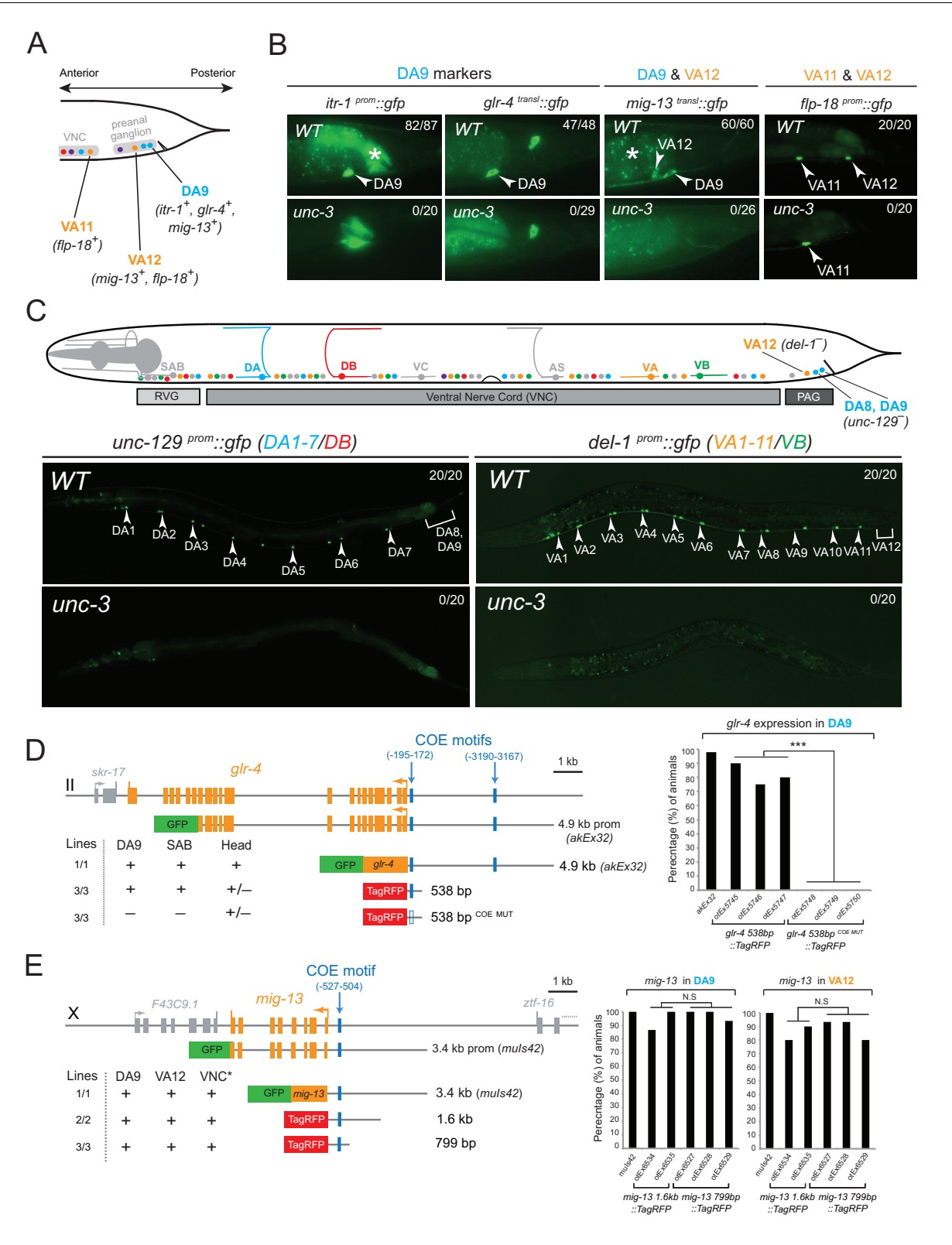

**Figure 2.** The terminal selector UNC-3 is required for MN subclass diversification. (**A**) Schematic showing subclass-specific gene expression in posterior MNs. (**B–C**) The expression of multiple subclass-specific markers (DA9: *itr-1, glr-4, mig-13*; VA12: *mig-13, flp-18*; DA1-7/DB: *unc-129*; VA1-11/VB: *del-1*) is dramatically affected in *unc-3(e151)* null mutant animals. Note that *flp-18* expression in VA11 is unaffected in *unc-3* null animals. Translational fusion reporters were used for *glr-4* (*glr-4* translation::gfp) and *mig-13* (*mig-13* translation::gfp), while promoter fragments were fused to *gfp* for *itr-1, flp-18, unc-129,* and

*Figure 2 continued on next page*

*Figure 2 continued*

*del-1*. Asterisks in **B** mark autofluorescence from the *C. elegans* intestinal cells. Numbers at top right of each image indicate the fraction of animals that showed marker expression in the respective MNs. Arrowheads in **B** and **C** mark MN subclasses (DA9, VA12, DA1-7 and VA1-11). In schematic shown in **C**, only DA, DB, VA, and VB MNs are color-coded. (**D–E**) *Cis*-regulatory mutational analysis is shown for *glr-4* (expressed in DA9 and SAB neurons) and *mig-13* (expressed in DA9 and VA12) genes. The *glr-4* and *mig-13* loci are schematized. Lines indicate genomic region fused to *gfp* (green) or *tagrfp* (red). (+) indicates robust reporter gene expression in the respective neurons. (±) indicates significant reduction in the number of neurons expressing the reporter when compared to transgenic animals carrying longer genomic fragments of the *cis*-regulatory region. (–) indicates complete loss or very faint expression in the respective neurons. Multiple transgenic lines were analyzed for each construct. At least twenty animals were analyzed per transgenic line. Detailed quantification of this analysis is provided on the right. Location of each COE motif (blue vertical lines) is presented as distance from ATG (+1). MUT indicates that COE motif has been mutated (shown as empty blue box) through substitution of 2 nucleotides in the core sequence (for example, COE wild-type site: T**CC**CNNGGGA >> COE MUT site: T**GG**CNNGGGA). ***p value < 0.001. Asterisk (*) on VNC in E indicates additional *mig-13* expression in VNC MNs anterior to the vulva.

*lin-39* and *mab-5* (*Figure 3B*). We conclude that – unlike *unc-3*, which is present in all MNs of a given class (*Figure 3A*). – the *C. elegans* Hox genes *lin-39, mab-5* and *egl-5* are expressed in a MN subclass-specific manner.

## *egl-5*, *lin-39* and *mab-5* are required for MN subclass diversification

We tested whether *lin-39* and *mab-5* (both expressed in DA neurons along the VNC) and *egl-5* (expressed in DA9) are, like *unc-3*, also required for the expression of MN subclass-specific genes. Given the early patterning defects that result from Hox gene removal in other organisms (*Duboule and Morata, 1994*; *McGinnis and Krumlauf, 1992*; *Wellik, 2007*; *Zakany and Duboule, 2007*), we first examined whether Hox genes are required for the generation or overall differentiation of cholinergic MNs. We find that Hox single or double null mutant animals show a normal number of cholinergic MNs (*Figure 4—figure supplement 1*). Moreover, Hox-deficient motor neurons still express pan-neuronal markers as well as genes that are shared by all cholinergic MNs (i.e. genes required for acetylcholine synthesis and vesicular packaging)(*Figure 4—figure supplement 1* and [*Stefanakis et al., 2015*]). However, we find defects in the expression of MN subclass-specific genes. Specifically, the expression of the DA1-7 marker *unc-129* is mildly affected in *lin-39* null animals, while *mab-5* mutants showed no significant effects (*Figure 4A*). Since *lin-39* and *mab-5* can function redundantly in *C. elegans* (*Liu and Fire, 2000*), we generated double mutants and found profound effects on *unc-129* expression (*Figure 4A*). Animals lacking *ceh-20*, the *C. elegans* PBX homolog known to function as Hox co-factor (*Mann and Chan, 1996*; *Mann et al., 2009*; *Merabet and Mann, 2016*; *Van Auken et al., 2002*), show similar effects to *lin-39 mab-5* double mutants (*Figure 4A*). Moreover, DA9 specific-genes (*itr-1, glr-4, mig-13*) are controlled by EGL-5, although the effects are milder compared to *unc-3* null mutants (*Figure 4C–E*). However, CEH-20 does not affect the expression of DA9 specific-genes (*Figure 4—figure supplement 1*), a finding reminiscent of previous reports showing that Abd-B proteins, like EGL-5, can function independently of PBX factors (*Rivas et al., 2013*; *Shen et al., 1997*; *van Dijk and Murre, 1994*). We conclude that Hox genes control the expression of subclass-specific genes within the DA neuronal class.

Consistent with their subclass-specific expression, we find that Hox genes are also required for subclass-specific effector gene expression in post-embryonically generated MN classes, such as the VA class (*Figure 1B*). *egl-5* is expressed in the VA12 neuron (located in the preanal ganglion) and not in other VA neurons along the VNC, and we find that the VA12-expressed genes *mig-13* and *flp-18* are under the control of *egl-5* (*Figure 4C–E*). Similar to the function of *lin-39* and *mab-5* in the more anteriorly, VNC-located DA neurons, we find that *lin-39 mab-5* double mutants, as well as *ceh-20/Pbx* null animals, affect the expression of *del-1*, a subclass-specific gene expressed in VA1-VA11 neurons, which is also under the direct control of UNC-3 (*Figure 4B*). We conclude that – similarly to the case of DA neurons – distinct Hox genes are expressed in subsets of VA neurons and together with UNC-3 generate VA subclass diversity.

Taken together, distinct Hox genes are expressed in a MN subclass-specific manner and – like UNC-3 – they control the expression of subclass-specific effector genes. Similarly to *unc-3*, expression of *lin-39, mab-5*, and *egl-5* in post-mitotic MNs persists into adulthood (*Figure 3B–C*), suggesting that Hox genes may act continuously and cell-autonomously to control MN subclass-specific

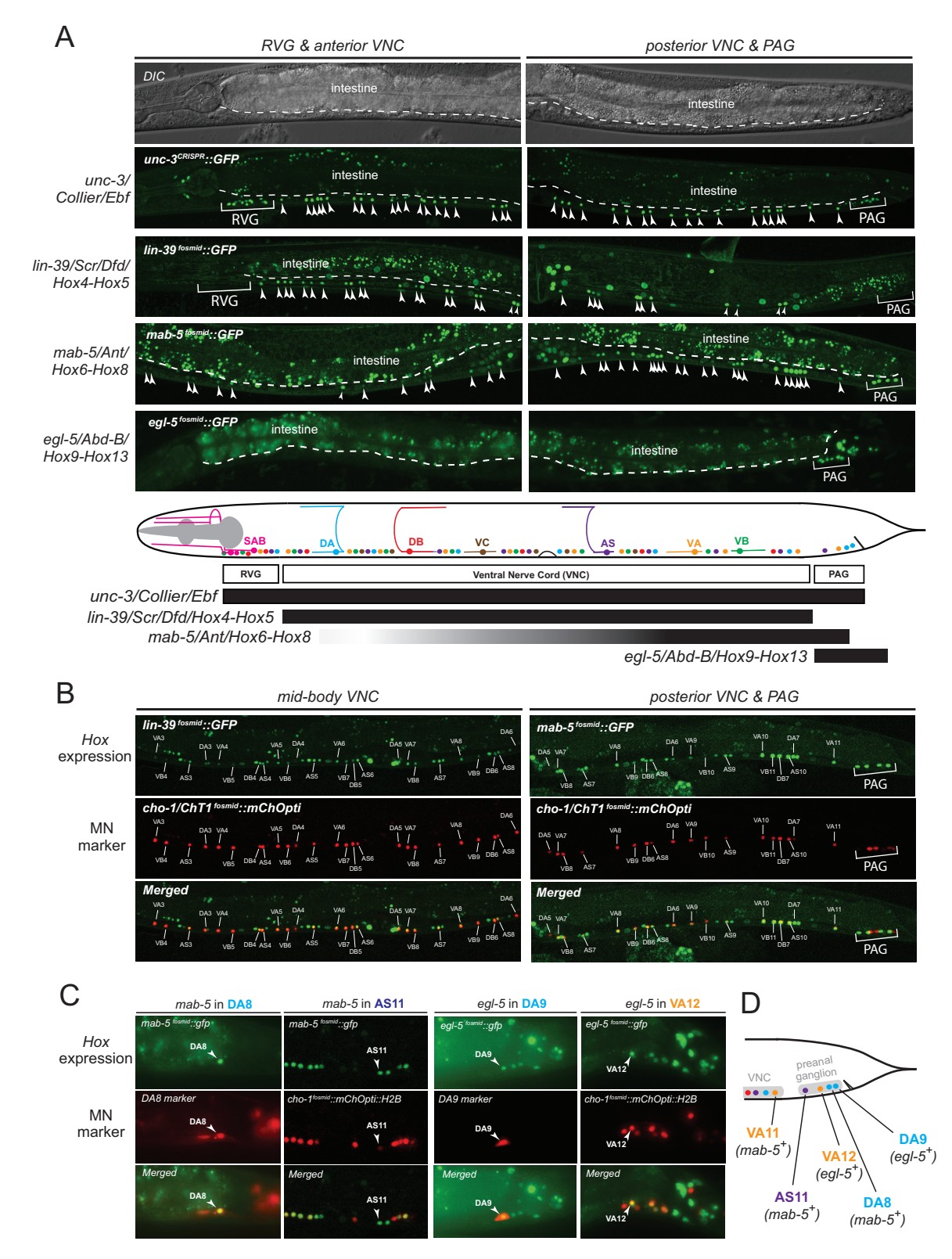

**Figure 3.** The *C. elegans* HOX genes *egl-5*/Abd-B/Hox9-Hox13, *lin-39*/Scr/Dfd. /Hox4-Hox5 and *mab-5* (Antp)/Hox6-Hox8 are expressed in a subclass-specific manner. (**A**) Fosmid-based reporters for *egl-5*, *lin-39* and *mab-5* reveal MN subclass-specific expression. A differential interference contrast (DIC) image of the anterior and posterior half of a *C. elegans* animal is provided at the top. This DIC image comes from the *unc-3*<sup>CRISPR</sup>*::gfp* animal, which is used to monitor endogenous *unc-3* expression. As previously reported (**Kratsios et al., 2011**; **Pereira et al., 2015**), *unc-3* is expressed in all

*Figure 3 continued on next page*

*Figure 3 continued*

individual neurons of a given MN class; expression is detected in MNs at the ganglia (RVG and PAG), as well as the ventral nerve cord (VNC). Arrowheads mark individual MN nuclei as expected since *gfp* is fused to the UNC-3, LIN-39, MAB-5, EGL-5 nuclear proteins. Autofluorescence of the intestinal cells is evident in the green channel. A dotted white line marks the limit of the intestinal cells, which are located dorsally to MNs. A schematic summarizing the intersectional expression of *unc-3* and *C. elegans* Hox genes in MNs is provided at the bottom. Hox gene names are provided with their fly and vertebrate paralogs. (B–C) Animals carrying fosmid-based Hox *gfp* reporters were crossed with animals carrying red fluorescent reporters for either all cholinergic MNs (*cho-1* ^fosmid^::*mChOpti*::*H2B*) or subclass-specific markers (DA8: *ser-2*^prom^::*mCherry*; DA9: *glr-4*^prom^::*tagrfp*). We found that Hox genes are expressed in a subclass-specific manner during larval and adult stages. Representative images of adult (day 1) animals are shown. (D) Schematic that summarizes HOX gene (*egl-5, mab-5*) expression in MNs of the preanal ganglion (PAG).

features in adult stages. Biochemical studies have indeed identified that LIN-39 binds onto its own *cis*-regulatory region (*Niu et al., 2011*), suggesting that it maintains its own expression through positive autoregulation.

## UNC-3 and Hox proteins act synergistically to generate MN subclass diversity

The effects on MN subclass-specific gene expression observed in *unc-3* and Hox mutants can be explained by *unc-3* and Hox participating either in linear or parallel genetic pathways. To distinguish between these two possibilities, we first monitored the expression of *egl-5, lin-39*, and *mab-5* fosmid-based reporters in *unc-3* null mutants. We did not detect any defects (*Figure 5A*). Conversely, the expression of *unc-3* is not affected in *egl-5* mutants (*Figure 5B*), suggesting that *unc-3* and Hox proteins may act in parallel pathways that function synergistically to control MN subclass-specific gene expression. The notion of independence is further supported by the lack of an effect of Hox genes on the expression of ACh pathway genes (*VAChT, ChAT, ChT*), which are direct UNC-3 target genes in all MNs (*Figure 4—figure supplement 1*).

If UNC-3 and Hox proteins act synergistically, we would predict that by generating double (*unc-3; egl-5*) or triple (*unc-3; lin-39 mab-5*) mutants, we would observe stronger effects compared to single mutants for *unc-3* and *egl-5* or double *lin-39 mab-5* mutants. Since animals carrying the *unc-3* null allele (*e151*) displayed severe effects (100% penetrance) in terms of MN subclass-specific gene expression (*Figure 2A–C*), we took advantage of a hypomorphic *unc-3* allele, *fp8* (*Richard et al., 2011*). While this allele shows defects as strong as the null allele for the DA9- and VA12-specific marker genes, its effects on *unc-129* (DA1-7 marker) and *del-1* (VA1-VA11 marker) are milder, thereby enabling combined mutant analysis. We find that *unc-3(fp8); lin-39 mab-5* triple mutants display a more severe effect on *unc-129* and *del-1* expression when compared to *lin-39 mab-5* double or *unc-3(fp8)* single mutants (*Figure 5C–D*). To sum up, the stronger effects on *unc-129* and *del-1* expression in *unc-3(fp8); lin-39 mab-5* triple mutants together with the absence of cross-regulatory interactions between UNC-3 and Hox lead us to conclude that these regulatory factors act synergistically to generate DA and VA subclass diversity by controlling the expression of subclass-specific genes. We summarize these findings into a genetic model shown in *Figure 5E*.

## UNC-3 and LIN-39 directly control *cis*-regulatory elements of subclass-specific genes

To elucidate the molecular basis for the synergistic function of UNC-3 and Hox genes, we considered the most parsimonious possibility of UNC-3 and Hox factors operating through distinct *cis*-regulatory motifs to activate subclass-specific effector genes. To this end, we took advantage of the modENCODE project which mapped transcription factor binding sites on an animal-wide level using ChIP-seq analysis (*Boyle et al., 2014*). Despite lacking cell-type specificity, these ChIP-seq studies have identified the GATTGATG sequence as a putative binding site for LIN-39 in *C. elegans* cells (*Figure 6A*). Focusing on two of the subclass-specific markers, *unc-129* and *del-1*, we found copies of this LIN-39 motif (as well as copies of the UNC-3 binding site, termed COE motif) in minimal *cis*-regulatory elements (243 bp for *unc-129* and 488 bp for *del-1*) that drive reporter gene expression in DA1-7/DB and VA1-11/VB MNs, respectively (*Figure 6B,E*). To test the functionality of the putative LIN-39 sites (GATTGATG), we mutated these sites in the *unc-129* element (243 bp ^LIN-39 SITES 6 & 7 MUT^) and observed a dramatic loss of reporter gene expression in the DA1-7 and DB neurons

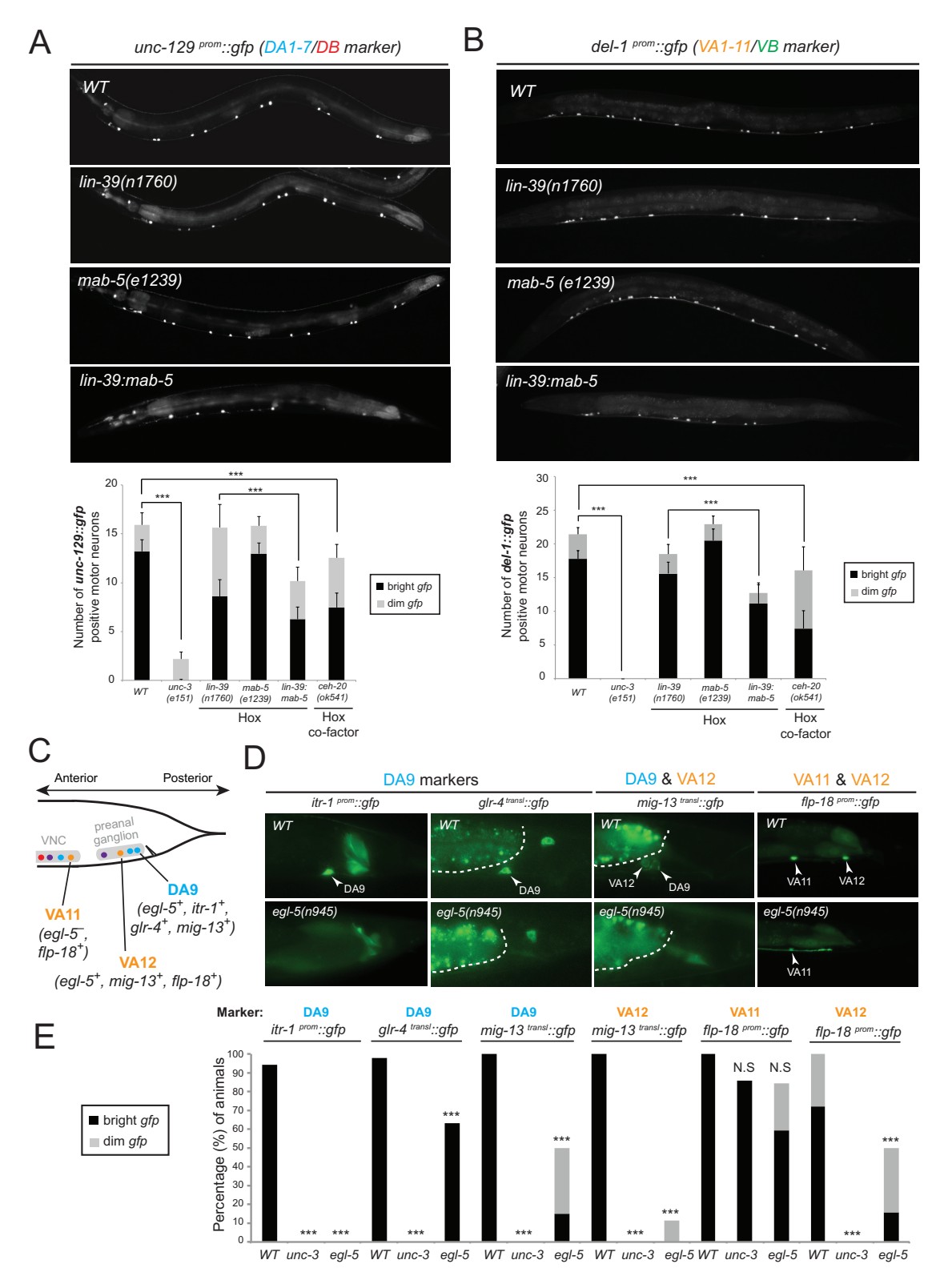

**Figure 4.** Hox genes – like *unc-3* – control the expression of MN subclass-specific genes along the A-P axis. (**A–B**) The expression of the subclass-specific markers *unc-129* (DA1-7/DB) and *del-1* (VA1-11/VB) is significantly affected in *lin-39 mab-5* double mutants, as well as in animals lacking *ceh-20*, *C. elegans* PBX homolog known to function as Hox co-factor. Quantification of the number of MNs expressing the *gfp* reporter (bright or dim) is provided at the bottom. Effects on reporter expression are milder in Hox null mutants when compared to *unc-3* null mutants. Error bars represent

*Figure 4 continued on next page*

Figure 4 continued

standard deviation (STDV). ***p value < 0.001. N > 20 animals. (C–D) The posterior Hox gene *egl-5*/Abd-B/Hox9-Hox13 controls subclass-specific genes (*itr-1, glr-4, mig-13, flp-18*) expressed in the most posterior member of the DA (DA9) and VA class (VA12). Effects on reporter expression are milder in *egl-5* null mutants when compared to *unc-3* null mutants. Similar effects were observed for multiple *egl-5* loss-of-function alleles (*n945, u202, tm4746*). A dotted white line marks the limit of the intestinal cells. (E) Quantification (percentage of animals) is provided with grey bars representing very dim *gfp* expression in the respective neuron, while black bars represent bright *gfp* expression. ***p value < 0.001. N > 20 animals.

The following figure supplement is available for figure 4:

**Figure supplement 1.** Analysis of acetylcholine pathway gene expression in Hox mutants and *ceh-20*/Pbx analysis.

(*Figure 6B–D*), which phenocopies the effect observed in *lin-39 mab-5* double mutants (*Figure 4A*). ChIP-seq data from the modENCODE consortium show adjacent binding for LIN-39 (as well as MAB-5) on the *unc-129_243* bp region that we have defined through our *cis*-regulatory analysis (*Figure 6—figure supplement 1*)(*Boyle et al., 2014*; *Niu et al., 2011*). Moreover, mutation of one LIN-39 site in the *del-1* element (488 bp $^{LIN-39\ SITE\ 4\ MUT}$) resulted in a reduction of reporter gene expression in VA1-11 and VB neurons (*Figure 6E–F*). Altogether, our findings suggest that, similar to UNC-3, the Hox protein LIN-39 (possibly together with MAB-5) directly controls expression of the MN subclass-specific genes *unc-129* and *del-1* through a discrete *cis*-regulatory motif (*Figure 5E*).

## Dual, homeotic activities of *egl-5/Abd-B*/Hox9-Hox13 in motor neuron subclasses

Mutations in Hox genes have traditionally been associated with the phenomenology of homeosis, which is the transformation of one body part into something that resembles another body part (*Bateson, 1894*). Previous work on *egl-5* null mutant animals indicated that the fates of specific tail blast (precursor) cells are transformed to those of their anterior homologues, suggesting homeotic transformation (*Chisholm, 1991*). To test this possibility in the context of the DA and VA neurons, we assessed the expression of the DA1-7 marker *unc-129* in DA9 of *egl-5* null mutants, as well as *del-1* (VA1-VA11 marker) in VA12 of *egl-5* mutants. We found that 100% of *egl-5* mutants display *del-1* de-repression in VA12, while in 84.2% of *egl-5* mutants *unc-129* is de-repressed in DA9 (*Figure 7A*). Hence, *egl-5* promotes posterior MN subclass identity and represses more anterior MN subclass identity.

Since EGL-5 is required for repressing *unc-129* in DA9, we also performed a sufficiency experiment where we misexpressed EGL-5 in DA1-7 neurons along the VNC and indeed found that EGL-5 is sufficient to repress *unc-129* expression in these neurons (*Figure 7B*). As the promoter (*unc-4*) we used to drive *egl-5* expression is also expressed in DA9, we observed partial rescue of *unc-129* de-repression in DA9 of *egl-5* null animals, indicating that EGL-5 acts cell autonomously (*Figure 7B*). We conclude that EGL-5 has a dual role in DA9 and VA12 neurons. Apart from working together with UNC-3 to activate the expression of DA9- and VA12-specific genes (*Figure 4C–E*, model in *Figure 5E*), EGL-5 is also required to repress features associated with DA1-7 and VA1-11 neurons located more anteriorly (*Figure 7A*).

Next, we sought to uncover the mechanistic basis of the homeotic transformation observed in DA9 and VA12 neurons. Since we have shown that *lin-39* and *mab-5* are not expressed in DA9 or VA12 but are both required for *unc-129* expression in DA1-7 and *del-1* expression in VA1-11, we reasoned that the gain of *unc-129* expression in DA9 and *del-1* expression in VA12 could be attributed to *lin-39* and/or *mab-5* expression being de-repressed in DA9 and VA12 neurons of *egl-5* mutant animals. We find no evidence for *mab-5* de-repression (*Figure 7—figure supplement 1*), but we observed *lin-39* de-repression in the DA9 and VA12 neurons in 100% of *egl-5* mutant animals (*Figure 7C*). In flies, worms and mice, Hox genes specifying posterior structures or neurons often repress the expression and activity of more anterior Hox genes (*Duboule and Morata, 1994*; *Gaufo et al., 2003*; *Hafen et al., 1984*; *Harding et al., 1985*; *Jung et al., 2010*; *Karlsson et al., 2010*; *Mallo et al., 2010*; *Schneuwly et al., 1987*; *Zheng et al., 2015*). Consistent with these reports, our analysis demonstrates, with single-neuron resolution, that the posterior Hox gene *egl-5* represses the mid-body Hox gene *lin-39* in DA9 and VA12 neurons.

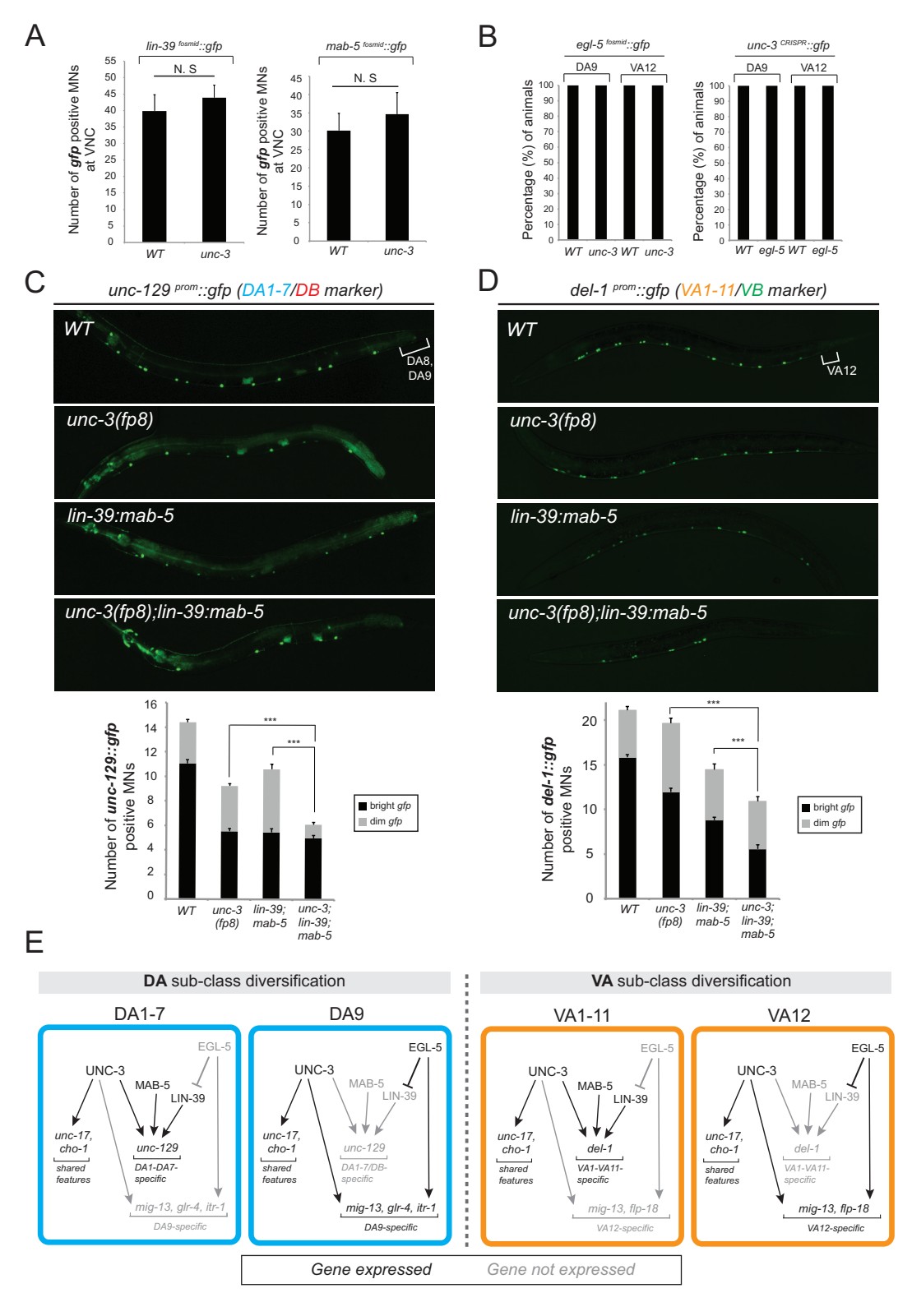

**Figure 5.** The terminal selector *unc-3* and the HOX genes *egl-5*/Abd-B/Hox9-Hox13, *lin-39*/Scr/Dfd/Hox4-Hox5 and *mab-5*/Antp/Hox6-Hox8 act synergistically. (**A**) The expression of *lin-39* and *mab-5* is not affected in ventral cord MNs of *unc-3* mutants. The number of *lin-39* and *mab-5* positive MNs was quantified in wild-type and *unc-3(e151)* animals. N.S: not statistically significant differences. Error bars represent standard deviation (STDV). N = 15. (**B**) The percentage of animals that show *egl-5* expression in DA9 and VA12 neurons of *unc-3(e151)* mutants was quantified and no differences

*Figure 5 continued on next page*

*Figure 5 continued*

were observed. Similarly, no differences in the expression of *unc-3* in the DA9 and VA12 neurons of *egl-5(n945)* mutants were observed. N = 10. (**C–D**) Animals carrying a hypomorphic *unc-3* allele (*fp8*) show mild defects on *unc-129* (DA1-7/DB marker) and *del-1* (VA1-11/VB marker) expression that are comparable to *lin-39 (n1760) mab-5 (e1239)* double mutants. Triple mutant animals *unc-3(fp8); lin-39 (n1760) mab-5 (e1239)* display even stronger (additive) effects providing evidence that *unc-3, lin-39* and *mab-5* act synergistically. Quantification of the number of neurons expressing the *gfp* reporter is provided. Error bars represent standard deviation (SEM). ***p value < 0.001. N = 20. Animals from all genotypes shown here were grown at 15°C because *fp8* is a hypomorphic, temperature-sensitive allele. (**E**) Genetic model to summarize our data for DA and VA subclass diversification.

We also performed the converse experiment. Since *egl-5* represses *unc-129* in DA9 and *del-1* in VA12, the loss of *unc-129* and *del-1* expression in DA and VA neurons along the VNC of *lin-39 mab-5* double mutants (*Figure 4A–B*) could be attributed to gain of *egl-5* expression in DA and VA neurons along the VNC. However, we found no differences when we monitored *egl-5* expression in MNs of *lin-39 mab-5* animals (*Figure 7—figure supplement 1*). To summarize, our findings suggest homeotic transformations on a single neuron level and are consistent with previous Hox studies describing the transformation of posterior neuronal identities to an anterior fate (*Gaufo et al., 2003*; *Jung et al., 2010*; *Karlsson et al., 2010*; *Zheng et al., 2015*). In animals lacking the posterior Hox gene *egl-5/Abd-B*/Hox9-Hox13, not only do the DA9 and VA12 neurons lose their subclass-specific features, but they also gain subclass-specific features of their anterior homologues due to de-repression of the mid-body Hox gene *lin-39/Dfd*/Hox4-Hox5.

## *egl-5/Abd-B*/Hox9-Hox13 controls synaptic wiring of the DA9 neurons

We further probed the extent of the homeotic subclass identity transformation in *egl-5* mutants by examining anatomical features of the transformed neurons. A defining feature of all DA neurons is that their axon migrates circumferentially to reach the dorsal nerve cord (DNC) where it runs anteriorly to form *en passant* neuromuscular synapses with dorsal body wall muscle (*Figure 8A*). In wild-type animals, DA9 forms neuromuscular synapses within a specific subaxonal domain (*Figure 8A*). We find that loss of *egl-5* results in misplacement of the DA9 neuromuscular synapses such that they are located more anteriorly in the DNC, in a territory normally reserved for neuromuscular synapses of more anteriorly located DA neurons (*Figure 8A*). These defects can be partially rescued by driving *egl-5* expression specifically in DA9 using the *glr-4* promoter which is partially affected by *egl-5* (*Figure 4D*), indicating that EGL-5 functions cell-autonomously to control aspects of DA9 identity (*Figure 8A*).

Apart from examining the synaptic output of DA9 neurons to muscle, we also examined the synaptic input to the DA9 neurons. DA9 receives sexually dimorphic synaptic inputs. In adult *C. elegans* males, DA9, but not the more anterior DA subclass members, is innervated by the AVG interneurons (*Jarrell et al., 2012*)(*Figure 1B*) and this selective input does not exist in DA9 of adult hermaphrodites. These synapses can be visualized with split GFP reporter technology ('GRASP') (*Feinberg et al., 2008*; *Oren-Suissa et al., 2016*). We find that in adult *egl-5* mutants, this synaptic input is lost, consistent with the neuron adopting the identity of a more anterior DA neuron (*Figure 8B*). The initial establishment of the synaptic input is, however, unaffected (L1 stage in *Figure 8B*), suggesting that *egl-5* controls maintenance of subclass-specific synaptic connectivity. Taken together, our analysis therefore demonstrates that a Hox gene not only controls molecular markers, but may also control MN subclass-specific synaptic wiring patterns.

## The Antennapedia-type Hox gene *mab-5* is required for DA, VA and AS subclass diversification

The data described above suggest that in *egl-5* mutants DA9 identity is transformed to that of its anterior homologs (anterior DA neurons). Since our expression analysis revealed that the Antennapedia-type Hox gene *mab-5* is expressed in DA8 and not in DA9 (*Figure 3C–D*), we examined whether in *mab-5* mutants DA8 might also transform into the fate of its anterior homologs. We indeed found that – similarly to the effect of EGL-5 in DA9 neurons – the expression of the DA1-7 marker *unc-129* is de-repressed in DA8 neurons of *mab-5* mutants (*Figure 9A–B*). DA8 not only adopts features of the more anterior DA neurons, but, unexpectedly, it also adopts features of the more posterior DA9 neuron. Specifically, the DA9-specific markers *mig-13, glr-4* and *itr-1* are de-repressed in the DA8

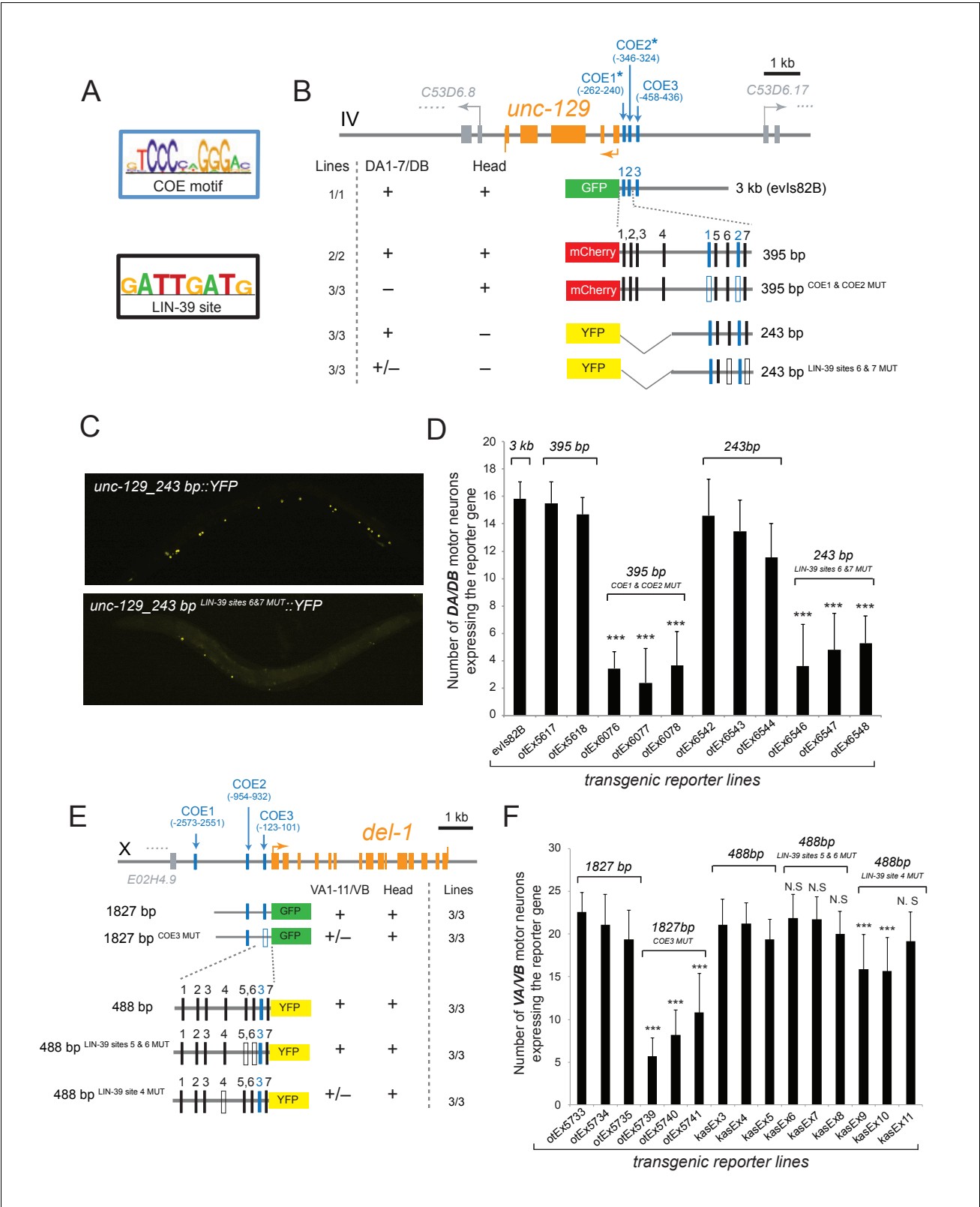

**Figure 6.** The Hox gene *lin-39*/Scr/Dfd/Hox4-Hox5 and the terminal selector *unc-3* regulate directly the expression of the subclass-specific genes *unc-129*/TGFβ and *del-1*. (**A**) The binding site for UNC-3 (COE motif) and LIN-39 are shown. The COE motif has been previously characterized (**Kratsios et al., 2011**), while the LIN-39 site was identified though ChIP-seq experiments (**Boyle et al., 2014**). (**B–D**) A schematic of the *unc-129* locus is shown. *Cis*-regulatory mutational analysis was performed in the context of transgenic animals. Lines indicate genomic region fused to

*Figure 6 continued on next page*

*Figure 6 continued*

*gfp* (green), *mCherry* (red) or *yfp* (yellow). (+) indicates robust reporter gene expression in DA1-7/DB and head neurons. (±) indicates significant reduction in the number of neurons expressing the reporter. (–) indicates complete loss or very faint expression in the respective neurons. Multiple transgenic lines were analyzed for each construct. At least 20 animals per transgenic line were analyzed. Detailed quantification of this analysis is provided in **D**. Location of each COE motif (UNC-3 binding site, shown as blue vertical line) is presented as distance from ATG (+1). The blue asterisk indicates that the COE motif is conserved in at least three other nematode species. The 395 bp and the 243 bp fragments contain two well-conserved COE motifs, which are required for expression in DA1-7/DB neurons. The 395 bp fragment contains 7 LIN-39 sites (1–7, represented as black vertical lines), while the 243 bp fragment contains 3 LIN-39 sites (5-7), two of which are required for reporter gene expression in DA1-7/DB neurons. MUT indicates deletion of the LIN-39 site or that the COE motif has been mutated through substitution of 2 nucleotides in the core sequence (for example, COE wild-type site: T<u>CC</u>CNNGGGA >> COE MUT site: T<u>GG</u>CNNGGGA). Empty blue or black boxes indicate that the COE motif and LIN-39 sites are mutated, respectively. Representative images of animals carrying *unc-129_243 bp::YFP* and *unc-129_243 bp* ^LIN-39 sites 6 and 7 MUT^::YFP are shown in **C** and detailed quantification is provided in **D**. Error bars represent standard deviation (STDV). ***p value < 0.001. N > 20 per transgenic line. (**E–F**) *Cis*-regulatory analysis of the *del-1* locus. The COE3 motif in the 1827 bp fragment is required for expression in VA and VB neurons, while we have previously shown that mutation of the COE2 does not affect expression in VA and VB neurons (*Kratsios et al., 2015*). The 488 bp fragment is sufficient to drive reporter gene expression in VA and VB neurons. Apart from the COE3 motif, the 488 bp fragment contains 7 LIN-39 sites (1–7, represented as black vertical lines) that are predicted with p value < 0.005. Unlike the combined deletion of LIN-39 sites 5 and 6, deletion of the LIN-39 site 4 (*del-1_488 bp* ^LIN-39 site 4 MUT^::YFP) results in a statistically significant loss of reporter gene expression in VA and VB neurons in two out of three transgenic lines. Detailed quantification is provided in F. Error bars represent standard deviation (STDV). ***p value < 0.0001. N > 20 per transgenic line.

The following figure supplement is available for figure 6:

**Figure supplement 1.** The Hox genes *lin-39/Scr/Dfd/*Hox4-Hox5 and *mab-5* (Antp) bind directly to the *cis*-regulatory region of MN subclass-specific genes.

neuron of *mab-5(-)* animals (*Figure 9A–B*)(*Sym et al., 1999*). ChIP-sequencing results from the mod-ENCODE project demonstrate that *mab-5* binds to the *cis*-regulatory region of *mig-13* and *itr-1* (*Figure 6—figure supplement 1*), indicating that *mab-5* may directly repress the expression of these DA9-specific genes in the DA8 neuron. Taken together, our results show that *mab-5* ensures the adoption of DA8 fate by preventing the expression of DA1-7-specific and DA9-specific features (*Figure 9C*). Hence, in the absence of *mab-5*, the DA8 neuron adopts a 'mixed' fate characterized by DA1-7 and DA9 traits.

Apart from its role in DA subclass diversity, we investigated whether *mab-5* is involved in subclass diversification of other MN classes, specifically, the VA and AS classes. *mab-5* is the only Hox gene to be expressed in VA11 and AS11 (*Figure 3B–D*). We find that *mab-5* is required for *avr-15/GluR* expression in VA11 and AS11 neurons (*Figure 9D–E*). *unc-3* is also required for *avr-15/GluR* (*Figure 9D–E*), thereby again corroborating the theme of *unc-3* cooperating with Hox genes in motor neuron subclass specification.

## *unc-3* and HOX also collaborate to control expression of intermediary regulatory factors

The data described above indicates a direct collaboration of the UNC-3 terminal selector with Hox genes in specific motor neuron subclasses on the *cis*-regulatory level of 'nuts and bolts' effector genes that define specific cellular phenotypes. We also found evidence for a collaboration of UNC-3 with HOX genes in controlling the expression of intermediary regulatory factors. Our previous work demonstrated the existence of a cohort of MN class-specific repressor proteins that are expressed in distinct motor neuron classes to antagonize UNC-3 protein function, generating diversity between different MN classes ('interclass diversity')(*Kerk et al., 2017*). The expression of these repressor proteins is not only restricted to individual MN classes, but within individual MN classes, some of these repressors are not uniformly expressed. For example, the ARID-type transcription factor *cfi-1* is expressed in the DA1-DA8 MNs, but not in DA9 (*Figure 1B*). *cfi-1* expression is controlled by *unc-3* (*Kerk et al., 2017*) and we investigated the impact of HOX genes on the subclass specificity of *cfi-1* expression. We found that the DA9-expressed gene *egl-5* is not involved in restricting *cfi-1* to DA1-8, but rather *lin-39* and *mab-5* are jointly required to activate *cfi-1* in DA1-DA8 (*Figure 9—figure supplement 1*). The loss of *cfi-1* expression in *lin-39 mab-5* mutants has the expected impact on *cfi-1* target gene expression. The ionotropic glutamate receptor *glr-4*, normally expressed only in the SAB neurons and the DA9 subclass neuron, becomes derepressed in the remaining DA neurons

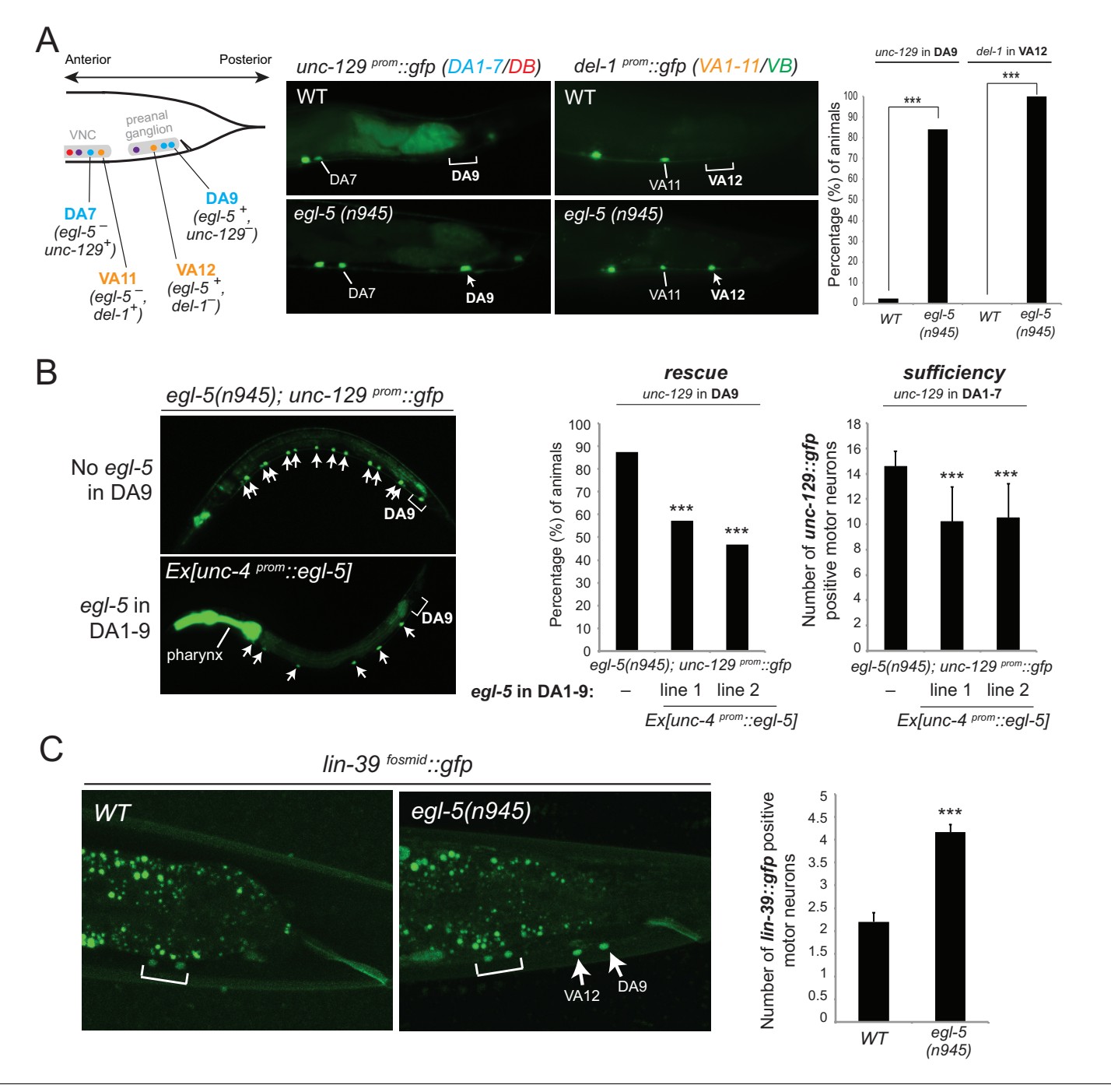

**Figure 7.** Evidence for homeotic activities of *egl-5*/Abd-B/Hox9-Hox13 in motor neuron subclasses. (**A**) Schematic showing that the subclass-specific markers *unc-129* (DA1-7/DB) and *del-1* (VA1-11/VB) are not expressed in DA9 and VA12 of WT animals, respectively. In *egl-5(n945)* mutants, *unc-129* is de-repressed in the DA9 neurons, while the expression of *del-1* is de-repressed in VA12 neurons. DA9 and VA12 were identified based on their axonal trajectory and cell body position. Quantification (percentage of animals) is provided on the right. ***p value < 0.001. N > 25. (**B**) Mis-expression of *egl-5* in all 9 DA neurons (DA1-9) (using a fragment of the *unc-4* promoter, *Ex [unc-4^prom^::egl-5]*) results in partial rescue of *unc-129* de-repression in DA9 of *egl-5* mutants. Quantification for *egl-5* rescue (percentage of animals) is provided on the right. ***p value < 0.001. In addition, mis-expression of *egl-5* in all 9 DA neurons results in repression of *unc-129* expression in anterior DA neurons (DA1-7), demonstrating that EGL-5 is also sufficient to repress *unc-129* expression. The number of DA1-7 and DB neurons that show *gfp* expression was quantified in two different transgenic lines *Ex [unc-4^prom^::egl-5] line 1(kasEx1)* and *2(kasEx2)*. Transgenic animals were identified based on the co-injection marker (*myo-2^prom^::gfp*), which drives *gfp* expression in the *C. elegans* pharynx. Error bars represent standard deviation (STDV). N > 15. (**C**) The expression of *lin-39* is de-repressed in DA9 and VA12 of *egl-5(n945)* mutants. White bracket indicates two VNC MNs that express *lin-39*. Quantification is provided on the right. ***p value < 0.001. N > 10.

*Figure 7 continued on next page*

*Figure 7 continued*

The following figure supplement is available for figure 7:

**Figure supplement 1.** Analysis of potential Hox cross-regulatory interactions.

(DA1-DA8) in *cfi-1* mutants (**Kerk et al., 2017**) and, consequently, also in *lin-39 mab-5* mutants (**Figure 9—figure supplement 1**). *cfi-1* regulation by *lin-39* and *mab-5* is HOX gene- and cell-specific, since the posterior HOX gene *egl-5* does not affect *cfi-1* expression, i.e. we observed no de-repression of *cfi-1* in the DA9 neuron of *egl-5* mutants (**Figure 9—figure supplement 1**). There is no cross-regulatory relationship between *cfi-1* and HOX genes, since HOX gene expression is unaffected in *cfi-1* mutants (**Figure 9—figure supplement 1**).

A similar effect of HOX genes can also be observed on the expression of another MN class-specific repressor, *bnc-1*. This Zn finger transcription factor is expressed in VA/VB MNs, is a direct target of UNC-3, and antagonizes the activity of UNC-3 on a number of effector genes in VA/VB (**Kerk et al., 2017**). *bnc-1* is expressed in all VA and VB neurons, except the anterior-most VB neuron VB1. This regionalized expression again suggests an impact of HOX genes on *bnc-1* expression. We indeed found that in *lin-39 mab-5* mutants, *bnc-1* expression is diminished in MNs in the midbody region (**Figure 9—figure supplement 1**). In conclusion, *unc-3* and HOX genes collaborate not only on the regulation of terminal effector genes that define terminal cellular phenotypes, but they also collaborate to regulate the expression of intermediary regulatory factors.

## Discussion

We provide here insights into the problem of how anatomically and functionally defined neuron classes diversify into specialized subclasses. Although the molecular mechanisms that generate interclass diversity, that is, differences between neuron classes, are beginning to be uncovered in multiple distinct systems, the mechanisms of subclass diversification have remained more elusive. In vertebrate and invertebrate nervous systems, numerous studies have identified transcription factors as key regulators of neuronal class specification by controlling the expression of class-specific features (**Allan and Thor, 2015**; **Hobert, 2011**). Whether and how these transcription factors are also required for subclass diversity has been less clear. Using *C. elegans* cholinergic VNC MN classes as a model, we describe here an intersectional gene regulatory strategy for MN subclass diversification along the A-P axis of the nervous system. First, we find that the terminal selector-type transcription factor UNC-3 – apart from controlling features shared by all MN class members – is also required for subclass diversification by regulating the expression of MN subclass-specific genes. Second, we demonstrate that three out of six *C. elegans* Hox genes (*lin-39, mab-5, egl-5*) are expressed in subsets of neurons of the DA, VA, and AS classes, and thereby differentially intersect with *unc-3*, which is expressed in all DA, VA, and AS neurons. These three Hox genes act synergistically and cooperate with UNC-3 in an intersectional manner to control the expression of MN subclass-specific genes along the A-P axis (**Figure 10**). Third, we uncovered that the posterior Hox gene *egl-5/Abd-B/*Hox9-Hox13, apart from acting as an UNC-3 co-activator to generate MN subclass diversity, is also required to repress the expression of the mid-body Hox gene *lin-39/Scr/Dfd/*Hox4-Hox5, while the Hox gene, *mab-5/Antp/*Hox6-Hox8, may operate as both an activator and repressor directly on its effector genes. Hence, the intersectional activity of a single, broadly-acting terminal selector and distinct Hox genes may constitute a conceptually straightforward solution to the intriguing problem of how neuronal subclass diversity is generated along the A-P axis of the nervous system.

Recent work has demonstrated that diversification of distinct *C. elegans* MN classes (i.e. interclass diversification) involves class-specific direct repression of effector genes (**Kerk et al., 2017**). At the mechanistic level, the ability of the terminal selector UNC-3 to activate MN class-specific genes is counteracted by transcriptional repressor proteins that are expressed in a MN class-specific manner. Hence, MN class-specific effector genes integrate positive regulatory input (through UNC-3) and negative regulatory input (through class-specific repressors). In contrast, our genetic and *cis*-regulatory analyses show that MN subclass diversity ("intraclass diversity") is mainly generated through positive regulatory inputs provided by UNC-3 and Hox proteins to subclass-specific effector genes

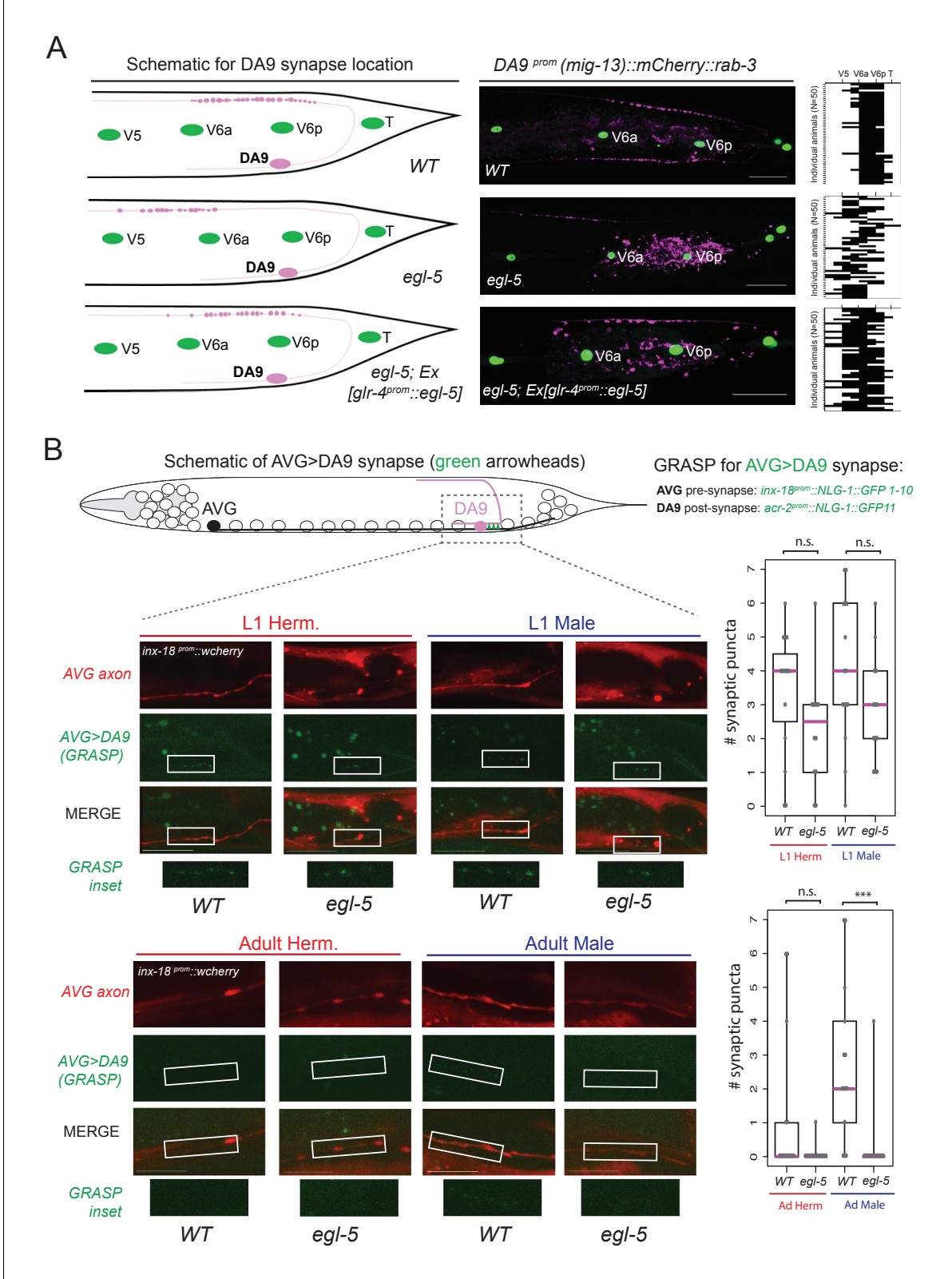

**Figure 8.** *egl-5*/Abd-B/Hox9-Hox13 affects synaptic wiring of the DA9 neurons. (**A**) Schematic showing the location of DA9 neuromuscular synapses in WT, *egl-5(n945)*, *egl-5(n945); Ex [glr-4prom::egl-5]* animals. On the right, the pre-synaptic boutons of the DA9 synapses were visualized using *mig-13 prom::mCherry::rab-3*. Since *mig-13* expression is partially affected in DA9 by *egl-5* (**Figure 4E**), we quantified the location of presynaptic boutons only in *egl-5* mutant animals in which *mig-13* expression was unaffected by the absence of *egl-5*. To quantify the location of the DA9 neuromuscular synapses

*Figure 8 continued on next page*

*Figure 8 continued*

(data provided on the right of each image), the seam cells (V5, V6a, V6p, T) were used as a reference since their location is not affected in *egl-5(n945)* mutants. It is noted that the gene promoter used for rescue (*glr-4*) is also under the partial control of *egl-5* in DA9 (*Figure 4E*). For the heat maps shown on the right N = 50. Each row represents a single animal. P value = 0.0238 for *egl-5(n945)* and *egl-5(n945); Ex [glr-4$^{prom}$::egl-5]* comparison. (**B**) Schematic of AVG>DA9 synapse, which is male-specific in the adult stage (see also *Figure 1B*). Synaptic inputs from AVG to DA9 were visualized with GRASP in *C. elegans* males and hermaphrodites at the first larval (L1) and adult stages and found to be defective in *egl-5(n945)* mutant male adult animals. The *otEx6342 (inx-18$^{prom}$::nlg-1::GFP1-10, acr-2$^{prom}$::NLG-1::GFP11, inx-18$^{prom}$::wCherry)* transgene was used to visualize the AVG>DA9 synapse with *gfp* and the AVG axon with *wCherry*. Quantification of fluorescent micrographs was performed by quantifying the number of synaptic puncta observed using GRASP (AVG>DA9) in L1 and adult hermaphrodites and males. ***p<0.001; n.s., not significant. Magenta horizontal bars represent the median, black boxes indicate quartiles, black lines indicate range. Each gray dot represents one quantified animal. Region of observed synaptic puncta marked with white boxes. Scale bar, 10 μm.

(*Figure 10*). By analogy to the terminal selector function of UNC-3 that controls shared (*Kratsios et al., 2015*, *2011*), class-specific (*Kratsios et al., 2015*, *2011*) and subclass-specific (this study) features of MNs throughout life, Hox proteins can be considered as 'sub-selectors' because they selectively regulate MN subclass-specific features. This regulatory principle of co-activation of subclass-specific genes is certainly not the only mechanism to achieve neuronal subclass diversity. For example, the diversification of the SAB class of head MNs and ASE class of gustatory neurons in *C. elegans* relies on direct repression of subclass-specific effector genes (*Etchberger et al., 2009*; *Kerk et al., 2017*).

Much of our current understanding of how Hox genes impact neuronal development comes from studies on mouse MNs in the hindbrain and spinal cord, as well as leg MNs, neuroblasts and peptidergic ventral nerve cord neurons in Drosophila (*Baek et al., 2013*; *Catela et al., 2016*; *Estacio-Gómez and Díaz-Benjumea, 2014*; *Estacio-Gómez et al., 2013*; *Karlsson et al., 2010*; *Mendelsohn et al., 2017*; *Miguel-Aliaga and Thor, 2004*; *Moris-Sanz et al., 2015*; *Philippidou and Dasen, 2013*; *Suska et al., 2011*). In *C. elegans*, Hox genes have also been recently shown to diversify touch receptor neuron identity (*Zheng et al., 2015*). Collectively, this large body of work has established that Hox genes are required for neuronal diversity, cell survival, axonal pathfinding and circuit assembly. However, only a handful of Hox downstream targets have been identified in the nervous system to date and the mechanistic details of how Hox genes interact with other transcription factors during neuronal development remained ill-defined. Our findings provide novel insights on the molecular mechanisms employed by Hox genes to control neuronal diversity: (i) by identifying seven effector genes and two intermediary regulatory factors (*cfi-1* and *bnc-1*) as Hox downstream targets and (ii) by demonstrating that distinct Hox genes intersect with a broadly-acting, terminal selector-type transcription factor and act together at the *cis*-regulatory level of effector genes to generate MN subclass diversity. These genes encode proteins critical for MN function throughout life, such as ion channels and neurotransmitter receptors. Given the maintained expression of Hox genes in adult *C. elegans* MNs and mouse CNS (*Hutlet et al., 2016*), we surmise that Hox genes, apart from controlling the expression of MN subclass-specific genes during development, may also maintain effector gene expression or serve additional roles in adult life. We indeed uncovered an unexpected role for the posterior Hox gene *egl-5*/Abd-B/Hox9-Hox13 in adult animals, as *egl-5*/Abd-B/Hox9-Hox13 is required for the maintenance (not the initial formation) of the synaptic input that the DA9 MN receives from AVG interneurons. Apart from adding a temporal dimension to the function of *egl-5*/Abd-B/Hox9-Hox13, these findings also demonstrate that Hox genes are required for synaptic input received by MNs, extending previous reports on the requirement of Hox genes for MN to muscle connectivity (*Catela et al., 2015*, *2016*; *Friedrich et al., 2016*; *Hessinger et al., 2017*; *Philippidou and Dasen, 2013*).

One phylogenetically conserved aspect of our findings is the distinctive nature of the mechanism through which the posterior Hox gene *egl-5*/Abd-B/Hox9-Hox13 operates in post-mitotic neurons to confer MN subclass diversity. Apart from acting as an UNC-3 co-activator to control the expression of subclass-specific genes in posteriorly located MNs (DA9 and VA12), EGL-5 also represses the expression of *lin-39*/Scr/Dfd/Hox4-Hox5, a mid-body Hox gene required for the expression of subclass-specific genes in more anterior DA and VA neurons (*Figure 5E*). Supporting the generality of this dual mechanism of action, a recent study demonstrated a similar, dual role for EGL-5 in the

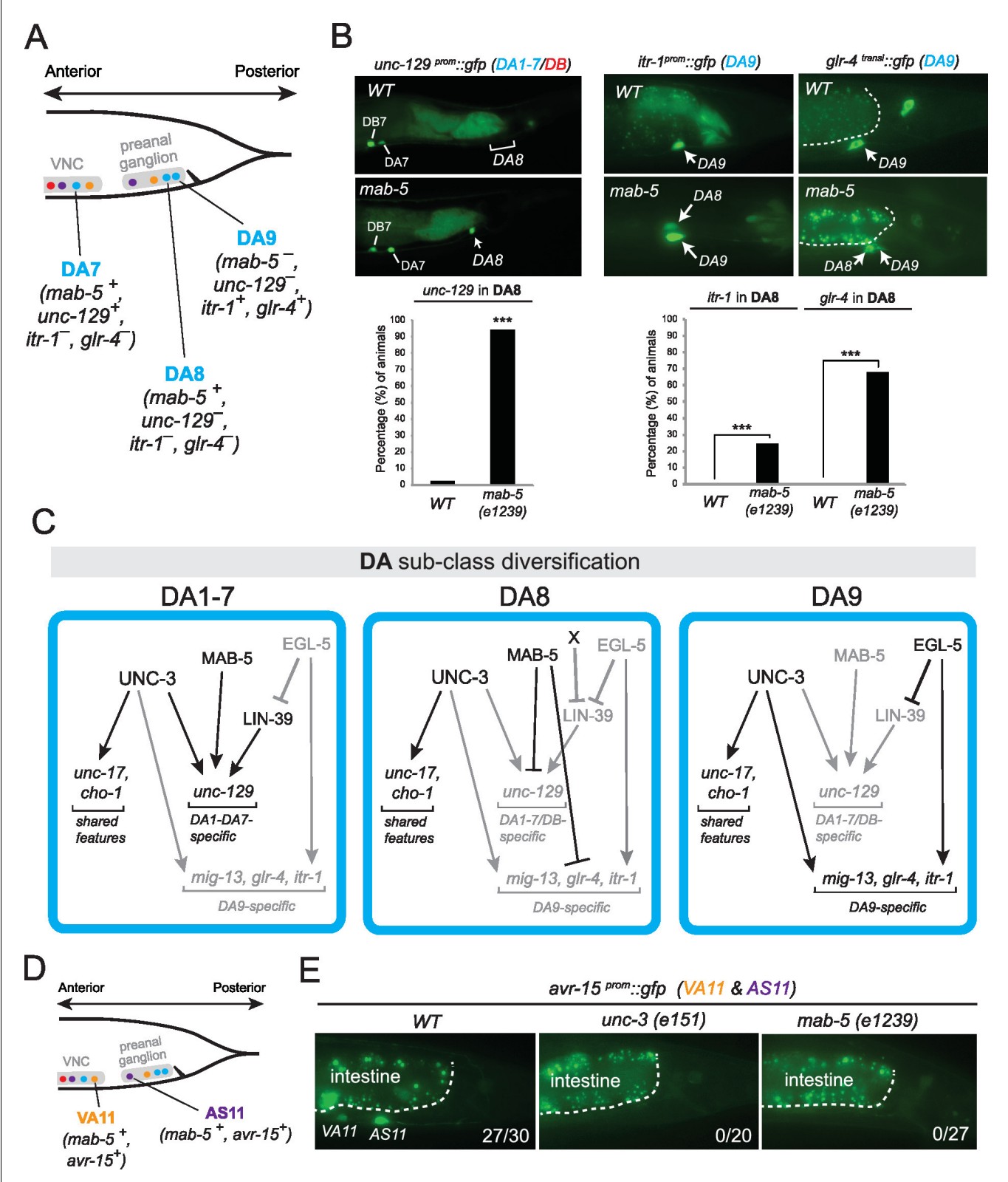

**Figure 9.** The Hox gene *mab-5*/Antp/Hox6-Hox8 is required for DA and AS subclass diversification. (**A**) Schematic summarizing the expression of *mab-5* and several subclass-specific markers (*unc-129, itr-1, glr-4*) in DA7, DA8 and DA9 neurons of WT animals. (**B**) The expression of the DA1-7/DB marker *unc-129* is de-repressed in DA8 neurons of *mab-5 (e1239)* animals. The WT image of *unc-129prom::gfp* is repeated from **Figure 7A**. The DA9 markers *itr-1* and *glr-4* are de-repressed in DA8 neurons of *mab-5 (e1239)* animals. Quantification (% of animals) is provided. DA8 was identified based on its

*Figure 9 continued on next page*

*Figure 9 continued*

axonal trajectory and cell body position. ***p value < 0.001. N > 30. (C) Genetic model that summarizes the function of *mab-5* in DA8 neurons, which appears distinct from the role of *mab-5* in DA1-7 neurons. (D) Schematic showing the expression of *mab-5* and *avr-15/GluR*, a subclass-specific gene expressed in VA11 and AS11, in posterior MNs. (E) The terminal selector *unc-3* and the Hox gene *mab-5* control *avr-15/GluR* expression in VA11 and AS11 neurons. Numbers at bottom right of each image indicate the fraction of animals that showed *avr-15/GluR* expression in VA11 and AS11. The white dotted line indicates the limits of the intestinal cells.

The following figure supplement is available for figure 9:

**Figure supplement 1.** *unc-3* and HOX also collaborate to control expression of intermediary regulatory factors.

diversification of posterior touch sensory neurons in *C.elegans* (*Zheng et al., 2015*). Moreover, the posterior Hox genes Hoxa10, Hoxc10, and Hoxd10 in the mouse spinal cord appear to function in a similar manner since they are required not only for the acquisition of posterior MN fate but also for repression of anterior MN identity (*Hostikka et al., 2009*; *Lin and Carpenter, 2003*; *Wahba et al., 2001*; *Wu et al., 2008*). This dual mechanism of posterior Hox gene action may explain how neuronal subclass diversity arises from an evolutionary standpoint, i.e. posterior Hox genes diversify posterior neurons from their more anterior counterparts. Supporting this possibility, homeotic transformations associated with Hox gene mutations have been proposed as a potential source of novelty at the single cell level that could contribute to the evolution of neuronal diversity (*Arlotta and Hobert, 2015*).

What are the upstream signals that establish Hox gene expression in specific MN subclasses? The signals that initiate *lin-39* and *mab-5* expression in VNC MNs are not clear, but it has been shown that epidermal growth factor (EGF) signaling from a neighboring cell to the P12 neuroblast (which produces the VA12 neuron) is necessary for the initiation of *egl-5* expression in P12 (*Jiang and Sternberg, 1998*). Such an intercellular signaling mechanism is the initiating event to diversify the P12 neuroblast and its descendant (VA12 neuron) from their anterior homologues. Since the EGF-like protein LIN-3 is expressed in various cells in the proximity of the posterior touch neuron neuroblast, it is conceivable that a similar mechanism may operate for the diversification of posterior touch neurons in *C.elegans* (*Zheng et al., 2015*).

Increasing evidence indicates that the intersectional strategy of distinct Hox genes and a broadly-acting transcription factor to generate neuronal subclass diversity (schematized in *Figure 10*) may be used in more complex nervous systems. We found that Ebf2, one of the four *unc-3* mouse orthologs, is co-expressed with distinct Hox genes in medial motor column (MMC) MNs along the A-P axis of the mouse spinal cord (*Figure 10—figure supplements 1* and *2*), leading us to speculate that Ebf2 and Hox may work together to generate MMC subclass diversity along the A-P axis, which is perhaps essential for innervation of distinct body wall (axial) muscles. Intriguingly, Hox genes and the transcription factor, FoxP1, are essential for diversification of spinal MNs of the lateral motor column (LMC) population, although the direct downstream effector genes of FoxP1 and Hox are largely unknown (*Dasen et al., 2008*; *Rousso et al., 2008*). Moving beyond MNs, a recent study on the molecular deconstruction of the serotonergic system in the mouse brainstem has revealed that the expression of distinct Hox genes intersects with the broadly-acting transcription factor Pet-1 in molecularly and functionally distinct populations of serotonergic neurons (*Okaty et al., 2015*). We propose that this intersectional gene regulatory principle may be utilized for the fractionation of neuronal classes into subclasses. Such fractionation is essential for the acquisition of subclass-specific traits, which could ultimately lead to the evolution of novel neuron types and circuits.

## Materials and methods

### *C. elegans* strains

Worms were grown at 20°C or 25°C on nematode growth media (NGM) plates seeded with bacteria (*E.coli* OP50) as a food source as previously described (*Brenner, 1974*). Mutant alleles used in this study: *unc-3 (e151) X, unc-3 (n3435) X, unc-3 (fp8) X, lin-39(n1760), mab-5(e1239), egl-5(n945), him-5*

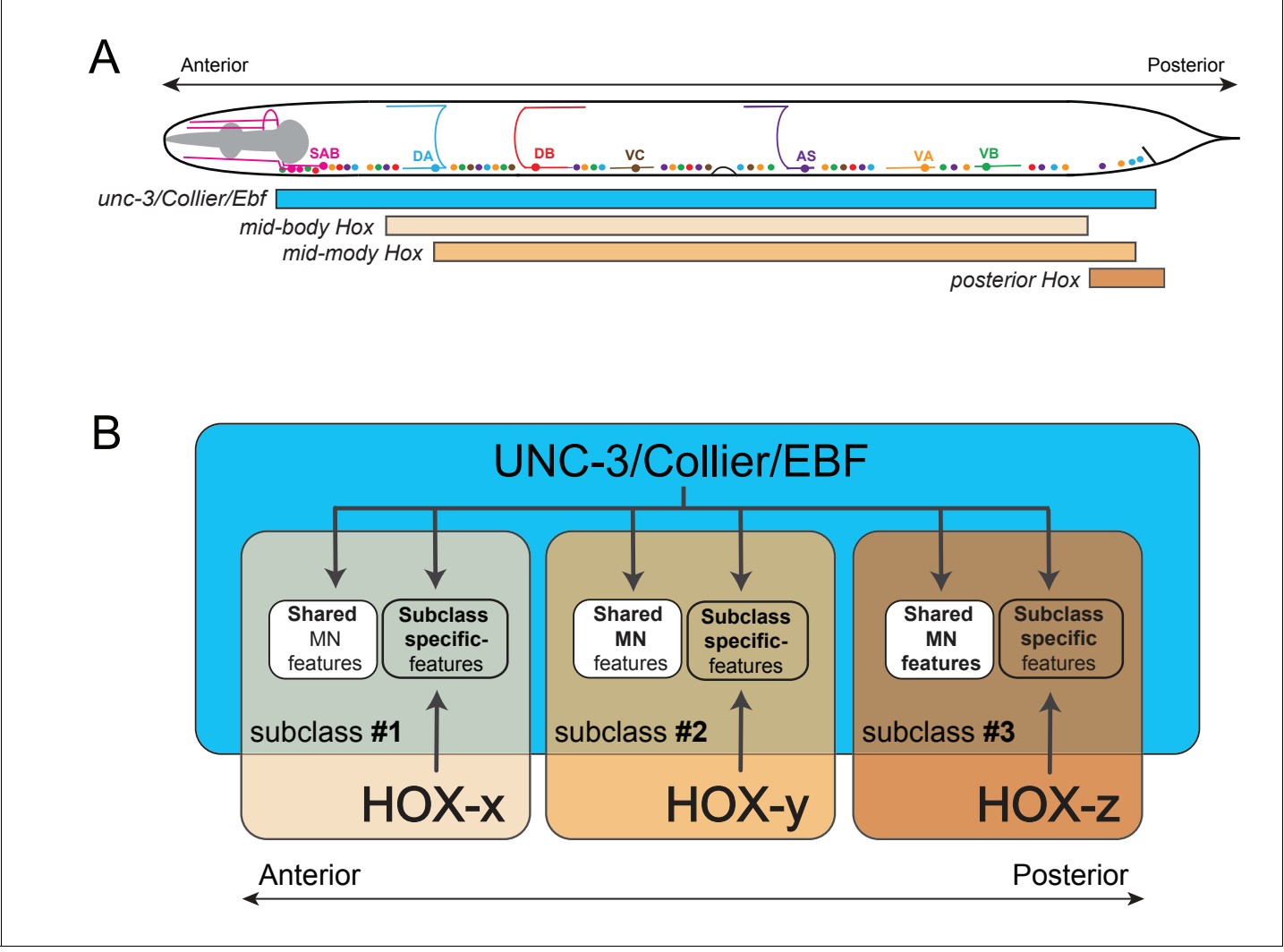

**Figure 10.** Model that conceptually summarizes the key findings of this paper. (**A**) A schematic summarizing the intersectional expression of *unc-3* and *C. elegans* Hox genes in MNs along the A-P axis. (**B**) Our findings reveal an intersectional strategy in which a non-regionally restricted, MN class selector gene (*unc-3/Collier/Ebf*) cooperates with a region-specific transcriptional code (HOX-x, HOX-y, HOX-z) to control the expression of MN subclass-specific features, providing a simple solution to the intriguing problem of how neuronal subclass diversity is generated along the A-P axis of the nervous system. Blue box indicates *unc-3/Collier/Ebf* expression in all MN classes along the A-P axis. Expression of distinct Hox genes (HOX-x, HOX-y, HOX-z) is presented with colored boxes of different shades of orange. The area of the intersection of *unc-3/Collier/Ebf* and Hox expression represents distinct MN subclasses (#1, #2, #3).

The following figure supplements are available for figure 10:

**Figure supplement 1.** The *unc-3* ortholog Ebf2 is co-expressed with distinct Hox genes in spinal motor neurons.

**Figure supplement 2.** Expression analysis for *Ebf1* and *Ebf3* in mouse spinal motor neurons.

(e1490), egl-5 (tm4746), egl-5(u202), ceh-20(ok541). bnc-1 (ot845 [bnc-1::+mNG+AID]) CRISPR allele, unc-3 (ot839 [UNC-3::+GFP]) CRISPR allele.

All reporter strains used in this study are: wdIs3 X or wdIs6 II [del-1$^{prom}$::gfp], evIs82B IV [unc-129$^{prom}$::gfp], mdIs123 [hum-2$^{prom}$::gfp], adEx1299 [avr-15$^{prom}$::gfp], ynIs59 [flp-18$^{prom}$::gfp], otIs107 [ser-2$^{prom1}$::gfp], muIs42 [mig-13$^{prom}$::MIG-13::gfp], otIs476 [glr-4$^{prom}$::tagrfp], otIs453 [itr-1$^{prom}$::gfp], sEx11976 [sra-36$^{prom}$::gfp], otIs544 [cho-1$^{fosmid}$::mChOpti::H2B], zfIs8 [ser-2$^{prom}$::mCherry], wgIs18 [lin-39$^{fosmid}$::gfp], wgIs27 [mab-5$^{fosmid}$::gfp], wgIs54 [egl-5$^{fosmid}$::gfp], akEx32 [glr-4$^{transl}$::gfp],

otEx5745-7 [glr-4$^{538bp\ prom}$::TagRFP], otEx5748-50 [glr-4$^{538bp\ prom\ COE\ mut}$::TagRFP], otEx6534-5 [mig-13$^{1.6kb\ prom}$::TagRFP], otEx6527-9 [mig-13$^{799bp\ prom}$::TagRFP], vsls48 [unc-17$^{prom}$::gfp], otEx5617-8 [unc-129$^{395bp\ prom}$::mChOpti], otEx6076-8 [unc-129$^{395bp\ prom\ COE1\ \&\ COE2\ MUT}$::mChOpti], otEx6542-4 [unc-129$^{243bp\ prom}$::YFP], otEx6546-8 [unc-129$^{243bp\ prom\ LIN-39\ sites\ 6\ \&\ 7\ MUT}$::YFP], otEx6342 (inx-18::nlg-1::GFP1-10), pEVL194 (acr-2::NLG-1::GFP11), inx-18p::wCherry; pRF4), kasEx1 [Punc-4$^{prom}$::egl-5 + myo-2::gfp], kasEx2 [Punc-4$^{prom}$::egl-5 + myo-2::gfp], wyIs226 [mig-13$^{prom}$::mCherry::rab-3], mxIs28[ceh-20$^{prom}$::ceh-20::yfp], mizEx57 [glr-4$^{prom}$::egl-5], otEx6502[cfi-1$^{fosmid}$::gfp], juIs14[acr-2$^{prom}$::gfp], kasEx3-5 [Pdel-1$^{488bp\ prom}$::NLS-YFP], kasEx6-8 [Pdel-1$^{488bp\ prom\ LIN-39\ sites\ 5\ \&\ 6\ MUT}$::YFP], kasEx9-11 [Pdel-1$^{488bp\ prom\ LIN-39\ site\ 4\ MUT}$::YFP], otEx5733-5 [Pdel-1$^{1827bp\ prom}$::GFP], otEx5739-41 [Pdel-1$^{1827bp\ prom\ COE3\ MUT}$::GFP], kasEx12[cfi-1$^{prom}$::tagRFP].

## Generation of transgenic reporter animals

Reporter gene fusions for *cis*-regulatory analysis described in *Figures 2* and *6* were made using a PCR fusion approach (*Hobert, 2002*). Genomic fragments were fused to *gfp*, or *yfp*, or *tagrfp* coding sequence, which was followed by the *unc-54 3' UTR*. The TOPO XL PCR cloning kit was used to introduce the PCR fusion fragments into the pCR-XL-TOPO vector (Invitrogen, Waltham, Massachusetts). Mutagenesis was performed using the Quickchange II XL Site-Directed Mutagenesis Kit (Stratagene, San Diego, California). PCR fusion DNA fragments were injected into young adult *pha-1 (e2123)* hermaphrodites at 50 ng/μl using *pha-1* (pBX plasmid) as co-injection marker (50 ng/μl).

## Microscopy

Worms were anesthetized using 100 mM of sodium azide (NaN$_3$) and mounted on 5% agarose on glass slides. Images were taken using an automated fluorescence microscope (Zeiss, AXIO Imager Z1 Stand) or Zeiss confocal microscope (LSM880). Acquisition of several z-stack images (each ~1 μm thick) was performed with the ZEN or Micro-Manager software (Version 3.1) (*Edelstein et al., 2010*). Representative images are shown following max-projection of 2–10 μm Z-stacks using the maximum intensity projection type. Image reconstruction was performed using Image J software (*Schneider et al., 2012*).

## Immunofluorescence staining

Mouse embryos were harvested between embryonic day 13 (e13.5) and e14.5, fixed in 4% paraformaldehyde for 1.5–2 hr, and processed for immunohistochemistry. Primary antibodies used: guinea pig anti-Hoxc6, mouse anti-Hoxc8, rabbit anti-Hoxc9, guinea pig anti-Hoxc10, and rabbit anti-Lhx3. Antibodies were provided by the lab of Thomas Jessell. Confocal images were obtained with a Leica SP5 confocal microscope and analyzed with ImageJ software.

## RNA in situ hydridization

Mouse embryos were harvested at embryonic day 14 (e14.5), fixed in 4% paraformaldehyde for 2 hr, and processed for in situ hybridization as previously described (*Dasen et al., 2005*). For the Ebf2 probe, we used the following primers:

5'-AAATTGTGGCCACTTTCTGG-3' (forward) and 5'-CCACAAAGTCCACGAAGGCT-3' (reverse). For Ebf1, Ebf3, and Ebf4 RNA in situ, we used previously published probes (*Garel et al., 1997*),

## Motor neuron subclass identification

Motor neuron subclasses were identified based on combinations of the following factors: (i) co-localization with or exclusion from another reporter transgene of known class-specific or subclass-specific expression, (ii) stereotypic position, either along the ventral nerve cord/within the retrovesicular or preanal ganglion, or relative to other motor neuron classes/subclasses, (iii) axonal trajectory, (iv) motor neuron birth order, and (v) number of neurons that belong to each MN class.

## Bioinformatic analysis

To predict the binding site (COE motif) for the transcription factor UNC-3 in the *cis*-regulatory region of *mig-13*, *glr-4* and *unc-129*, we used the MatInspector program from Genomatix (http://www.genomatix.de) (*Carthalius et al., 2005*). The Position Weight Matrix (PWM) for the LIN-39 binding site is catalogued in the CIS-BP (Catalog of Inferred Sequence Binding Preferences)

database (*Weirauch et al., 2014* http://cisbp.ccbr.utoronto.ca/), allowing us to identify the LIN-39 site on the *unc-129* and *del-1 cis*-regulatory region using FIMO (Find Individual Motif Occurrences; [*Grant et al., 2011*]), which is one of the motif-based sequence analysis tools of the MEME (Multiple Expectation maximization for Motif Elicitation) bioinformatics suite (http://meme-suite.org/). For the FIMO analysis, the p-value threshold was set at <0.01 for *unc-129* and <0.005 for *del-1*.

## Statistical analysis

For data quantification, graphs show either values expressed as mean ± standard deviation (SD) or as percentage (%) of animals. When values are shown as mean ± standard deviation (SD), the statistical analyses were performed using the unpaired *t*-test (two-tailed). When values are shown as percentage (%), the statistical analyses were performed using Fisher's exact test (two-tailed). Calculations were performed using the GraphPad QuickCalcs online software (http://www.graphpad.com/quickcalcs/). Differences with p<0.001 were considered significant. For data shown in *Figure 8A*, a two-way ANOVA was performed. For data shown in *Figure 8B*, we performed non-parametric Mann–Whitney test (Wilcoxon rank sum test).

## Acknowledgements

We thank Chi Chen and Jihad Aburas for the generation of transgenic strains, the CGC, which is funded by NIH Office of Research Infrastructure Programs (P40 OD010440), for providing strains, James Rand and Stephen Fields for providing the *mdIs123* transgene, Tulsi Patel for providing the *unc-3*$^{CRISPR}$*::gfp* allele, Eviatar Yemini for communicating the expression pattern of *flp-18* in MN subclasses, Nicholas Kramer for initial analysis of synaptic phenotype of egl-5 mutants, Richard Mann, Ellie Heckscher, Edwin Ferguson, and members of the Hobert lab for comments on this manuscript. This work was funded by the Howard Hughes Medical Institute, the NIH (2R37NS039996), a K99 fellowship (K99NS084988) and a NINDS grant (R00NS084988) to P.K, and an HFSP Career Development Award and Tomizawa Jun-ichi & Keiko Fund to K.M. GGC's laboratory is supported by the Italian Telethon Foundation.

## Additional information

### Funding

| Funder | Grant reference number | Author |
|---|---|---|
| National Institute of Neurological Disorders and Stroke | R00NS08498 | Paschalis Kratsios |
| Fondazione Telethon | | G Giacomo Consalez |
| Human Frontier Science Program | Career Development Award and Tomizawa Jun-ichi & Keiko Fund | Kota Mizumoto |
| Howard Hughes Medical Institute | | Oliver Hobert |
| National Institute of Neurological Disorders and Stroke | R37NS039996 | Oliver Hobert |

The funders had no role in study design, data collection and interpretation, or the decision to submit the work for publication.

### Author contributions

PK, Conceptualization, Data curation, Formal analysis, Funding acquisition, Validation, Investigation, Visualization, Methodology, Writing—original draft, Project administration, Writing—review and editing; SYK, CC, KM, Formal analysis, Validation, Investigation, Visualization, Writing—review and editing; JL, BV, EAB, WF, EDDLC, LC, GGC, Investigation; OH, Conceptualization, Supervision, Funding acquisition, Project administration, Writing—review and editing

**Author ORCIDs**

Paschalis Kratsios, http://orcid.org/0000-0002-1363-9271

Oliver Hobert, http://orcid.org/0000-0002-7634-2854

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
