## [Decision Letter]

Thank you for submitting your article "An intersectional gene regulatory strategy defines intraclass diversity of *C. elegans* motor neurons" for consideration by *eLife*. Your article has been favorably evaluated by a Senior Editor and three reviewers, one of whom is a member of our Board of Reviewing Editors..

The reviewers have discussed the reviews with one another and the Reviewing Editor has drafted this decision to help you prepare a revised submission.

Summary:

The study by Kratsios et al. addresses the role of Hox transcription factors and COE factor Unc3 in the subtype diversification of cholinergic motor neurons in *C. elegans*. The authors characterize a set of markers that distinguish subclasses of motor neurons and show that Hox genes are required for their expression. They provide compelling evidence that Hox proteins act in conjunction with the transcription factor Unc3 to regulate expression of subclass specific genes. The authors also show that the posterior Hox protein *egl-5* establishes motor neuron subtype identity by repressing the expression of more anterior Hox genes.

While the reviewers find the conclusions of this study compelling, we were not completely convinced regarding the novelty. The role of *unc-3* in MN specification has been previously demonstrated by the authors. The role of Hox genes in neuronal subclass specification, including MN specification, has been extensively studied in several other model systems. The strength in this study lies in the identification of an extensive repertoire of subclass-specific MN effector genes, and the demonstration that these genes, as well as neuronal morphology and connectivity, are controlled by a COE-Hox code. In addition, the enhancer analysis, combined with ModEncode ChIP data, provides a much more elaborate molecular resolution for MN subclass specification than that described in other systems. Addressing the following concerns, in particular points 1 and 2 below, could help strengthen the manuscript.

Essential revisions:

1) In a recent study from the Hobert lab, the group identified seven transcription factors acting together with Unc-3 to control MN specification (PMID 28056346). It would be interesting to see if any of these regulators (*unc-4, mab-9, cfi-1, vab-7, unc-55, lin-13, bnc-1*) are regulated by the Hox homeotic genes studied in the current study, and vice versa. This would only need to be analyzed in the relevant cells i.e., *egl-5* vs. *cfi-1* in DA9, etc.

2) A recent study in *Drosophila* reveals that the Hox homeotic gene Ubx acts in both MNs and their target muscles to control the formation of neuromuscular connections (PMID 27913640). This touches upon the long-standing idea that TFs that are expressed both in neurons and the target tissues could coordinate axon-target pathfinding via activating the same surface molecules in both cells. On that note, can they rule out a role for *egl-5* in the target muscle as an underlying reason for the synapse location defects observed in these mutants (Figure 8)?

3) There are a few places where the authors imply that certain conceptual findings are novel, but have been well-characterized in the literature at a neuron-specific level. The transformation of posterior neuronal identities to an anterior fate, and in some cases the derepression of Hox genes, have been shown in flies, worms, and mice. These include the transformation of peptidergic neurons in fly Hox mutants (PMID:20485487), and the derepression of Hox1 genes in Hox3 mouse mutants (PMID:12954718), and the derepression of Hox genes in Hoxc9 mutants (PMID:

20826310). The authors should rephrase statements that imply novelty in cases where it has already been demonstrated at a neuronal subtype-specific level.

4) The reviewers do not believe the data showing expression of Hox proteins and Ebf2 in median motor column neurons in mice adds much to the paper. Hox genes are expressed by most motor neuron subtypes, and there is no current evidence that they are important for the development of this motor neuron subtype. The authors should remove this section from the main Results, perhaps mention it in the Discussion, and show it as a supplement.

5) One limitation of this analysis is the attempt to generalize the finding of Unc-3 controlling some motor neuron subtype genes to a general modus operandi. It is not clear whether Unc-3 motifs are overrepresented in all subtype specific promoters or a few were selected because of Unc-3 sites and ENCODE data.

6) Terminology: we understand the challenging complexity of the system, but the terminology still seems unnecessarily confusing. For example, in Figure 1 the authors use "subclass", "inter-class" and "intra-class". And "subclass" and "intra-class" comes up in other places in the manuscript. Could not intra-class be removed from the paper, and replaced with subclass. Of course, MNs constitute a subclass of neurons, DA:s a subclass of MNs, and DA9 a sub-type of DA:s. There is an example of a sentence that will surely confuse many readers: "However, the manner by which UNC-3 activity is restricted to individual subclasses is fundamentally distinct from the manner by which UNC-3 activity is restricted to distinct MN classes." Perhaps use "sub-type" and unique neuron when referring to DA9, and use subclass of DA:s and MNs, etc.

---

## [Author Response]

*[…] While the reviewers find the conclusions of this study compelling, we were not completely convinced regarding the novelty. The role of unc-3 in MN specification has been previously demonstrated by the authors. The role of Hox genes in neuronal subclass specification, including MN specification, has been extensively studied in several other model systems. The strength in this study lies in the identification of an extensive repertoire of subclass-specific MN effector genes, and the demonstration that these genes, as well as neuronal morphology and connectivity, are controlled by a COE-Hox code. In addition, the enhancer analysis, combined with ModEncode ChIP data, provides a much more elaborate molecular resolution for MN subclass specification than that described in other systems. Addressing the following concerns, in particular points 1 and 2 below, could help strengthen the manuscript.*

*Essential revisions:*

*1) In a recent study from the Hobert lab, the group identified seven transcription factors acting together with Unc-3 to control MN specification (PMID 28056346). It would be interesting to see if any of these regulators (unc-4, mab-9, cfi-1, vab-7, unc-55, lin-13, bnc-1) are regulated by the Hox homeotic genes studied in the current study, and vice versa. This would only need to be analyzed in the relevant cells i.e., egl-5 vs. cfi-1 in DA9, etc.*

We pursued this point by evaluating the expression of *cfi-1/Arid3a* and *bnc-1/Bnc1/2*, two out of the seven repressors for which we have generated reliable reagents to monitor and quantify their expression. In wild-type animals, *cfi-1* is expressed in several cholinergic (DA1-DA8, DB, VA, VB) and GABAergic (DD, VD) MN classes of the *C. elegans* ventral nerve cord, while *bnc-1* is selectively expressed in all 12 VA neurons, most VB neurons (not in VB1) and 2 of 3 SAB neurons (SABVL/R) (Kerk et al. Neuron, 2017, PMID: 28056346). In brief, we found that HOX genes indeed regulate the expression of these subtype-specific repressors, explained in detail in a new section on p.16/17 and in a new Figure 9—figure supplement 1. These findings nicely extend the collaboration of UNC-3 with HOX to another set of effector genes and therefore further expand the scope of the manuscript. We thank the reviewers for proposing us to do this; it further strengthened the manuscript.

*2) A recent study in Drosophila reveals that the Hox homeotic gene Ubx acts in both MNs and their target muscles to control the formation of neuromuscular connections (PMID 27913640). This touches upon the long-standing idea that TFs that are expressed both in neurons and the target tissues could coordinate axon-target pathfinding via activating the same surface molecules in both cells. On that note, can they rule out a role for egl-5 in the target muscle as an underlying reason for the synapse location defects observed in these mutants (Figure 8)?*

Following this suggestion, we assessed whether *egl-5* is also expressed in posterior body wall muscle and found this to be the case. We therefore performed an additional rescue experiment by providing EGL-5 in the muscle (using the *myo-3* promoter) of *egl-5* mutants and observed partial rescue of the DA9 synapse location phenotype. However, the muscle-specific rescue was less convincing than the neuronal DA9-specific rescue and adding the muscle-expressed construct to the neuronal expressed construct did not obviously improve rescue. Due to the inconclusive nature of these results, we would therefore prefer to refrain from raising these points in the manuscript.

*3) There are a few places where the authors imply that certain conceptual findings are novel, but have been well-characterized in the literature at a neuron-specific level. The transformation of posterior neuronal identities to an anterior fate, and in some cases the derepression of Hox genes, have been shown in flies, worms, and mice. These include the transformation of peptidergic neurons in fly Hox mutants (PMID:20485487), and the derepression of Hox1 genes in Hox3 mouse mutants (PMID:12954718), and the derepression of Hox genes in Hoxc9 mutants (PMID:*

*20826310). The authors should rephrase statements that imply novelty in cases where it has already been demonstrated at a neuronal subtype-specific level.*

We do not disagree with what the reviewers say, but want to emphasize that what makes our paper novel is the cellular and molecular resolution with which we characterize Hox gene function and its collaboration with UNC-3. Meaning, we show how HOX genes interact with terminal selectors at the cis-regulatory level, with the resolution of single targets and single cells.

As per the reviewers’ suggestion, we have modified the text in Results to acknowledge that our findings are consistent with previous studies in worms, flies and mice. As suggested, we cite these papers PMID:20485487, PMID:12954718, PMID: 20826310, PMID: 26539892 in two occasions in the Results section:

“In flies, worms and mice, Hox genes specifying posterior structures or neurons often repress the expression and activity of more anterior Hox genes (PMID:20485487, PMID:12954718, PMID: 20826310, PMID: 26539892 and others). Consistent with these reports, our analysis demonstrates, with single neuron resolution, that the posterior Hox gene *egl-5* represses the mid-body Hox gene *lin-39* in DA9 and VA12 neurons.

“To summarize, our findings suggest homeotic transformations on a single neuron level and are consistent with previous Hox studies describing the transformation of posterior neuronal identities to an anterior fate (PMID:20485487, PMID:12954718, PMID: 20826310, PMID: 26539892).”

Moreover, we have added a new paragraph in Discussion that discusses the conserved aspect of our findings regarding posterior Hox gene activity and neuronal diversity.

*4) The reviewers do not believe the data showing expression of Hox proteins and Ebf2 in median motor column neurons in mice adds much to the paper. Hox genes are expressed by most motor neuron subtypes, and there is no current evidence that they are important for the development of this motor neuron subtype. The authors should remove this section from the main Results, perhaps mention it in the Discussion, and show it as a supplement.*

We respectfully disagree with the respect to eliminate this data. As detailed above, we think one key novelty of our finding is integration of UNC-3/EBF and HOX gene function. To show that such integrated function may also be *possible* in vertebrates is a notion that we would like to display in the main text. Specifically, we believe that the expression of Ebf2 particularly in the median motor column (MMC) is an important finding that highlights the potential evolutionary implications of our *C. elegans* results. The *C. elegans* motor neurons in the ventral nerve cord and the two flanking ganglia (RVG, PAG) could perhaps be envisioned as an “ancient MMC” since these motor neurons innervate body wall muscles. There is indeed no current evidence in the mouse literature suggesting that Hox genes are important for the development/diversity of body wall muscle-innervating motor neurons. However, it is conceivable that this lack of evidence is most likely due to the lack of molecular markers that can subdivide MMC motor neurons into MMC subclasses along the A-P axis. On the other hand, the plethora of *C. elegans* motor neuron markers allowed us to reveal that Hox genes are required for the development and diversity of body wall muscle-innervating motor neurons.

*5) One limitation of this analysis is the attempt to generalize the finding of Unc-3 controlling some motor neuron subtype genes to a general modus operandi. It is not clear whether Unc-3 motifs are overrepresented in all subtype specific promoters or a few were selected because of Unc-3 sites and ENCODE data.*

UNC-3 motifs are indeed overrepresented in the subclass-specific gene promoters. Eleven out of the 12 subclass-specific gene promoters contain at least one UNC-3 (COE) motif. In addition, 9 out of 10 of the subclass-specific genes tested for UNC-3 dependency are indeed *unc-3* dependent as revealed by our mutant analysis. To help the reader, we have clarified this point in the Results (subsection “The terminal selector gene unc-3 is required for MN subclass diversity”) and have also added one extra column in Table 1 entitled “UNC-3 dependency and number of COE motifs”. In addition, we now provide experimental evidence that – apart from the subclass-specific gene *unc-129* – functional Hox (LIN-39) binding sites are also present in the *cis*-regulatory region of the subclass-specific gene *del-1*. This new data are now shown in Figure 6. Altogether, these findings indicate that the regulation of subtype-specific genes by UNC-3 and Hox may constitute a general modus operandi.

*6) Terminology: we understand the challenging complexity of the system, but the terminology still seems unnecessarily confusing. For example, in Figure 1 the authors use "subclass", "inter-class" and "intra-class". And "subclass" and "intra-class" comes up in other places in the manuscript. Could not intra-class be removed from the paper, and replaced with subclass. Of course, MNs constitute a subclass of neurons, DA:s a subclass of MNs, and DA9 a sub-type of DA:s. There is an example of a sentence that will surely confuse many readers: "However, the manner by which UNC-3 activity is restricted to individual subclasses is fundamentally distinct from the manner by which UNC-3 activity is restricted to distinct MN classes." Perhaps use "sub-type" and unique neuron when referring to DA9, and use subclass of DA:s and MNs, etc.*

Inspired by other examples in biology – inter-cellular vs. intra-cellular, inter-chromosomal vs. intra-chromosomal, etc. –, we attempted to use the terms inter-class vs. intra-class to refer to neuronal diversity. However, we recognize that the terminology can be confusing especially between intra-class and sub-class. As suggested, we have replaced throughout the manuscript and figures the term “intraclass” with “subclass” for clarity.

We have also modified the sentence "However, the manner by which UNC-3 activity is restricted to individual subclasses is fundamentally distinct from the manner by which UNC-3 activity is restricted to distinct MN classes." to "However, the manner by which UNC-3 activity is restricted to individual subclasses (members of a given class) is fundamentally distinct from the manner by which UNC-3 activity is restricted to distinct MN classes."